# Idiosyncrasies in Large Language Models

**Mingjie Sun** [* 1]    **Yida Yin** [* 2]    **Zhiqiu Xu** [3]    **J. Zico Kolter** [1]    **Zhuang Liu** [4]

## Abstract

In this work, we unveil and study idiosyncrasies in Large Language Models (LLMs) – unique patterns in their outputs that can be used to distinguish the models. To do so, we consider a simple classification task: given a particular text output, the objective is to predict the source LLM that generates the text. We evaluate this synthetic task across various groups of LLMs and find that simply fine-tuning text embedding models on LLM-generated texts yields excellent classification accuracy. Notably, we achieve 97.1% accuracy on held-out validation data in the five-way classification problem involving ChatGPT, Claude, Grok, Gemini, and DeepSeek. Our further investigation reveals that these idiosyncrasies are rooted in word-level distributions. These patterns persist even when the texts are rewritten, translated, or summarized by an external LLM, suggesting that they are also encoded in the semantic content. Additionally, we leverage LLM as judges to generate detailed, open-ended descriptions of each model's idiosyncrasies. Finally, we discuss the broader implications of our findings, including training on synthetic data, inferring model similarity, and robust evaluation of LLMs. Code is available at github.com/locuslab/llm-idiosyncrasies.

## 1. Introduction

As the adoption of generative models such as LLMs accelerates, it becomes increasingly important to understand the origin and provenance of such generated content. While a great deal of past work has focused on the classification of human-written and AI-written content (Krishna et al., 2023; Mitchell et al., 2023; Sadasivan et al., 2025), there has been little work on classifying *between* content generated by different LLMs, either between the outputs of entirely different models or between those of different variants of the same model family. If possible, the ability to distinguish between source models in this manner would be valuable for a number of applications: it could shed light on the relative uptake of different LLMs, beyond what is reported by individual companies, and on the nature of data used to build different models. Additionally, it could offer insights into what features of generated text are most "unique" to each LLM.

In this paper, we investigate whether LLMs exhibit idiosyncrasies that enable their outputs to be reliably differentiated. Inspired by recent studies on dataset bias in computer vision (Liu & He, 2025; Zeng et al., 2024), which showed that images from different large-scale datasets can be accurately distinguished by standard neural networks, we consider a similar synthetic classification task to assess the separability of responses generated between different LLMs. Specifically, we sample a large number of text outputs from each LLM using the same set of prompts and then train a classifier to recognize which model generates a specific text. Figure 1 provides an overview of our framework. The illustrated example on ChatGPT, Claude, Grok, Gemini, and DeepSeek presents a five-way classification problem.

We find that a classifier based upon simple fine-tuning text embedding models on LLM outputs is able to achieve remarkably high accuracy on this task. This indicates the clear presence of idiosyncrasies in LLMs. The observation is highly robust over a large variety of LLM combinations. For instance, trained on the combined set of texts from ChatGPT, Claude, Grok, Gemini, and DeepSeek, a model can achieve 97.1% classification accuracy on the *held-out* validation data, compared to a 20.0% chance-level guess. Within the same model family, we obtain a non-trivial 59.8% accuracy across 4 model sizes in Qwen-2.5 series (Qwen et al., 2024). Further, we observe strong out-of-distribution generalization of these classifiers when tested on responses from prompts outside the training distribution.

We observe several interesting properties of this task. When controlling the length and format of outputs through prompt instructions, we still obtain high classification accuracy. Furthermore, for post-trained LLMs, the classifier demonstrates non-trivial accuracy even with only the first few tokens of the generated text. However, when classifying generations

---

[*]Equal contribution [1]Carnegie Mellon University [2]UC Berkeley [3]University of Pennsylvania [4]Princeton University. Correspondence to: Mingjie Sun <mingjies@andrew.cmu.edu>, Yida Yin <davidyinyida0609@berkeley.edu>.

*Proceedings of the $42^{nd}$ International Conference on Machine Learning*, Vancouver, Canada. PMLR 267, 2025. Copyright 2025 by the author(s).

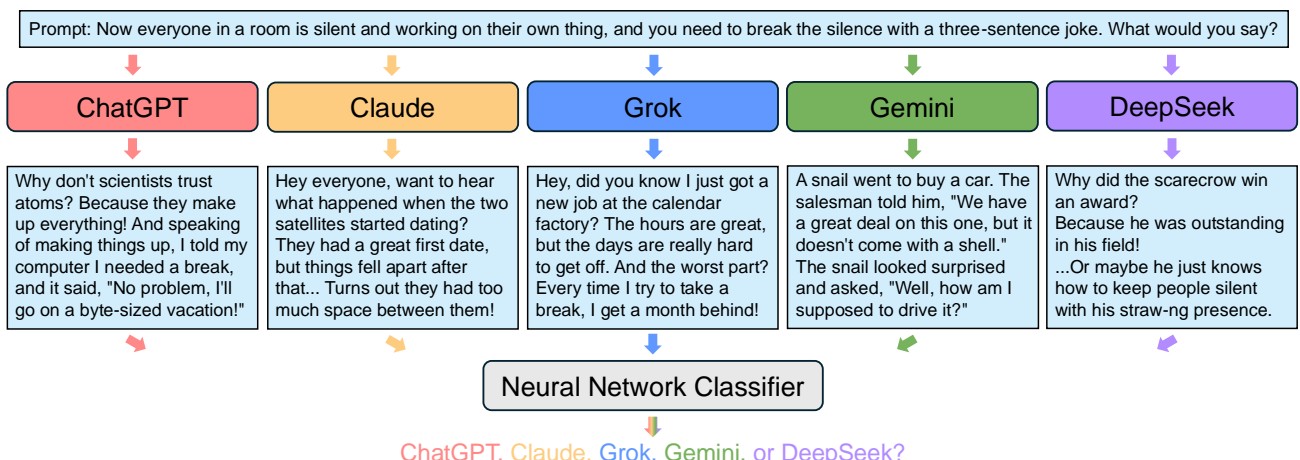

*Figure 1.* **Our framework for studying idiosyncrasies in Large Language Models (LLMs).** We show that each LLM is unique in its expression. In the example shown here on ChatGPT, Claude, Grok, Gemini, and DeepSeek, a neural network classifier is able to distinguish them with a near-perfect 97.1% accuracy.

from the *same* LLM but using different sampling strategies, we achieve accuracy only slightly above the chance level. In addition, we observe certain behaviors of this task that resemble those of standard text classification, where improvements in text embeddings and availability of larger training datasets lead to better classification performance.

We analyze the sources of these idiosyncrasies by applying text transformations that isolate different levels of information. We find that randomly shuffling words in LLM-generated responses leads to only a slight decrease in classification accuracy. This suggests that a substantial portion of distinctive features is encoded in the word-level distribution. We then highlight distinct sets of characteristic phrases that are consistently associated with each LLM. We also observe that markdown formatting contributes to a moderate degree of idiosyncrasies in the LLMs following post-training.

At the same time, we obtain over 90% accuracy when the word distribution is disrupted through transformations that preserve semantics, such as rephrasing or translating. Even with the most aggressive transformation – summarizing, classification accuracy remains well above chance-level guess. This finding implies that semantic information also shapes the idiosyncrasies in LLMs. Through open-ended language analysis, we provide further insights into these characteristics. For instance, ChatGPT has a preference for detailed, in-depth explanations, whereas Claude produces more concise and direct responses, prioritizing clarity.

Last, we discuss the broader implications of our findings. One should be cautious when using synthetic data to train LLMs, as we show that many of these idiosyncrasies can be inherited in such a process. Our framework also serves as a tool for assessing model similarities among frontier models, either open-source or proprietary. In addition, we discuss how the idiosyncrasies in LLMs can be used maliciously to

manipulate voting-based leaderboards, therefore highlighting the need for more robust evaluation methodologies.

## 2. Evaluating Idiosyncrasies in LLMs

Large Language Models (LLMs) share several core characteristics. The majority of them are based on the Transformer architecture (Vaswani et al., 2017), which is shared by all models we consider in this paper. Second, they are trained using an auto-regressive objective (Radford et al., 2019), where they predict the next token in a sequence based on preceding context. Lastly, their training datasets significantly overlap, often incorporating vast and diverse sources such as Common Crawl, Wikipedia, and Stack Overflow. Given these similarities, it is natural to ask: do LLMs speak in the same way? If not, how can we effectively measure the degree of their differences?

To address these questions, we construct a synthetic task focused on classifying outputs from different LLMs. Consider $N$ LLMs, denoted as $f_1, \ldots, f_N$, where each $f_i$ takes an input prompt $p$ and outputs a text completion $o$. For a given dataset $\mathcal{D}$ of prompts, the outputs produced by each LLM $f_i$ are denoted as $\mathcal{O}_i$. We approach this problem with a straightforward setup. For $N$ output sets $\mathcal{O}_i$, we formulate a $N$-way classification task, where the objective is to predict which LLM produced each output. If outputs of different LLMs were drawn from the same distribution, classification accuracy would not be better than random chance. Thus, we use the classification performance of this synthetic task as a measure of idiosyncrasies in LLMs.

Our task is formulated as a sequence classification problem, for which fine-tuning BERT-style models is a common approach (Sun et al., 2019). In this work, we fine-tune a more recent and competitive sequence embedding model based

| ChatGPT | Claude | Grok | Gemini | DeepSeek | acc. (chat) |
|---|---|---|---|---|---|
| ✓ | ✓ | | | | 99.3 |
| ✓ | | ✓ | | | 97.7 |
| ✓ | | | ✓ | | 98.7 |
| ✓ | | | | ✓ | 97.2 |
| | ✓ | ✓ | | | 99.7 |
| | ✓ | | ✓ | | 99.6 |
| | ✓ | | | ✓ | 99.6 |
| | | ✓ | ✓ | | 99.4 |
| | | ✓ | | ✓ | 98.7 |
| | | | ✓ | ✓ | 99.9 |
| ✓ | ✓ | ✓ | ✓ | ✓ | 97.1 |

(a) chat APIs

| Llama | Gemma | Qwen | Mistral | acc. (instruct) | acc. (base) |
|---|---|---|---|---|---|
| ✓ | ✓ | | | 99.9 | 98.3 |
| ✓ | | ✓ | | 97.8 | 81.7 |
| ✓ | | | ✓ | 97.0 | 96.3 |
| | ✓ | ✓ | | 99.9 | 98.3 |
| | ✓ | | ✓ | 99.9 | 98.4 |
| | | ✓ | ✓ | 96.1 | 95.7 |
| ✓ | ✓ | ✓ | ✓ | 96.3 | 87.3 |

(b) instruct and base LLMs

*Table 1.* **Classification accuracies for various LLM combinations.** *Top*: results for chat APIs. *Bottom*: results for instruct and base LLMs. Check marks (✓) denote the models included in each combination. We observe high classification accuracies consistently across all model combinations, indicating the presence of distinct idiosyncrasies in LLMs.

on decoder-only Transformers: LLM2vec (BehnamGhader et al., 2024). We attach a $N$-way classification head to the extracted embeddings and use LoRA-based fine-tuning (Hu et al., 2022) to the model weights. Input sequences are truncated to a maximum length of 512 tokens. We report the classification accuracy on a held-out validation set. Additional training details are provided in Appendix A.2.

## 2.1. Main Observations

We observe surprisingly high accuracies by neural networks to classify LLM outputs. This observation is robust across different settings, *e.g.*, across model families and sizes.

We describe the LLMs we use to generate the output datasets $\mathcal{O}_{1,\cdots,N}$. For a comprehensive and fair comparison across model families, we categorize three groups of LLMs:

1. Chat APIs ("chat"): This category includes state-of-the-art LLMs that are primarily accessible via APIs. We consider GPT-4o (OpenAI, 2024), Claude-3.5-Sonnet (Anthropic, 2024), Grok-2 (xAI, 2024), Gemini-1.5-Pro (Google, 2024), and DeepSeek-V3 (DeepSeek-AI, 2024). For simplicity, we refer to them as ChatGPT, Claude, Grok, Gemini and DeepSeek. Their architectures and weights remain proprietary and undisclosed, with the exception of DeepSeek.

2. Instruct LLMs ("instruct"): These models are trained to generate high-quality responses from human instructions. We consider four LLMs of similar sizes across different families: Llama3.1-8b (Dubey et al., 2024), Gemma2-9b (Riviere et al., 2024), Qwen2.5-7b (Qwen et al., 2024) and Mistral-v3-7b (Jiang et al., 2023). We will refer to them as Llama, Gemma, Qwen and Mistral.

3. Base LLMs ("base"): These are base versions of instruct LLMs. They are obtained by pretraining on extensive text corpora without any post-training stage.

Throughout the paper, we refer to these three categories as "chat", "instruct", and "base" respectively. For a given prompt dataset, we collect 11K text sequences, splitting them into 10K and 1K as training and validation sets, respectively. The same split is used across all LLMs. For chat APIs and instruct LLMs, we generate outputs from UltraChat (Ding et al., 2023), a diverse dialogue and instruction dataset. For base LLMs, we synthesize new texts using prompts from FineWeb (Penedo et al., 2024), a high-quality LLM pretraining dataset. More details on response generation are in Appendix A.1.

**Across model families.** In Table 1, we report the results for classifying outputs from various combinations of chat APIs (Table 1a) and instruct / base LLMs (Table 1b). In each

of the three LLM groups, we enumerate all ($C_N^2$) possible pairwise combinations when choosing 2 out of $N$ models in the top panel of each table, as well as the case including $N$ models in the bottom row. For the binary classification task, the neural network consistently achieves over 90% accuracy, with only one exception. Notably, for chat APIs and instruct LLMs, many combinations reach as high as 99% accuracy. In the more challenging $N$-way classification tasks, our classifiers maintain strong performance, achieving at least 87.3% accuracy across three groups. These results highlight the idiosyncrasies across different LLMs. We refer readers to Appendix B.3 for the confusion matrices of our classifiers.

**Within the same model family.** We evaluate sequence classification performance when distinguishing responses from LLMs within the same model family. Note that models from the same family typically share common training procedures, *e.g.*, pretraining datasets and optimization schedule. First, we analyze the impact of model size by considering four Qwen2.5 instruct LLMs with 7B, 14B, 32B, and 72B parameters. As shown in Table 2, the classification task is more difficult here, but our classifiers remain well above chance accuracy when distinguishing LLMs within the same family. In the binary classification setup, the highest accuracy reaches 85.5%, whereas in the full combination setup, the accuracy becomes 59.8%. In addition, we observe high accuracies when classifying responses from base and instruct versions of the same model. For example, our classifiers achieve 96.8% accuracy when distinguishing outputs from Qwen2.5-7b base and instruct models.

| 7b | 14b | 32b | 72b | instruct |
|----|-----|-----|-----|----------|
| ✓ | ✓ | | | 77.0 |
| ✓ | | ✓ | | 81.2 |
| ✓ | | | ✓ | 83.4 |
| | ✓ | ✓ | | 63.1 |
| | ✓ | | ✓ | 85.5 |
| | | ✓ | ✓ | 84.8 |
| ✓ | ✓ | ✓ | ✓ | 59.8 |

*Table 2.* **Classification within Qwen2.5 model family.** The classifier can differentiate responses between LLMs within the same model family with reasonably well accuracies.

**Generalization to out-of-distribution responses.** We find that our classifiers generalize robustly to responses beyond their training distribution. To evaluate this, we collect responses from instruct LLMs across four diverse datasets: *i.e.*, UltraChat, Cosmopedia (Ben Allal et al., 2024), LmsysChat (Zheng et al., 2024), and WildChat (Zhao et al., 2024). These datasets originate from different sources and are designed for various purposes – Cosmopedia is intended for synthetic data generation, LmsysChat and WildChat capture real-world user interactions, while UltraChat consists primarily of synthetic responses. For each dataset, we train a classifier on a group of model responses and evaluate the

classifier on the remaining three datasets. Table 3 shows the results on instruct LLMs. Our classifiers generalize well across different datasets, indicating that they learn very robust and transferable patterns.

| train / test | UltraChat | Cosmopedia | LmsysChat | WildChat |
|--------------|-----------|------------|-----------|----------|
| UltraChat | 96.3 | 98.9 | 89.9 | 92.4 |
| Cosmopedia | 95.7 | 99.8 | 88.3 | 94.9 |
| LmsysChat | 94.7 | 97.2 | 91.8 | 92.0 |
| WildChat | 95.1 | 99.1 | 90.2 | 95.7 |

*Table 3.* **Robust generalization to out-of-distribution responses.** We train classifiers on LLM outputs from one prompt dataset and tested on those from another.

### 2.2. Controlled Experiments

We analyze the behaviors of the synthetic classification task in several controlled settings. *From now on, we only report accuracies of the $N$-way classification task in each group.*

**Prompt-level interventions.** We assess the degree of idiosyncrasies in LLM outputs with explicit prompt-level interventions. Specifically, we modify the original prompt by incorporating additional instructions to constrain response length and format. We then perform sequence classification on the resulting outputs. Our interventions are:

- Length control: *Please provide a concise response in a single paragraph, limited to a maximum of 100 words.*
- Format control: *Please provide your response in plain text only, avoiding the use of italicized or bold text, lists, markdown, or HTML formatting.*

LLM outputs after these interventions are presented in Appendix C. We find that LLMs can follow the additional instructions in generating responses.

| | original | length control | format control |
|--|----------|----------------|----------------|
| instruct LLMs | 96.3 | 93.0 | 91.4 |

*Table 4.* **Controlling LLM outputs with prompts.** An instruction is added to the original prompt to specify the output length and format. *Length control* limits responses to one paragraph. *Format control* ensures that responses are in plain text without any format.

The results are shown in Table 4, where "original" means the classification accuracy without interventions. We can see that neural networks still perform excellently for classifying LLM outputs applied with length and format control prompts. These findings suggest that LLM characteristics are deeply embedded in the generated text, persisting despite surface-level constraints on length and formatting.

**Input length of text embedding models.** We control the number of input tokens to the text embedding models. Specifically, we truncate each response to a fixed number of tokens in a left-to-right fashion. Figure 3 presents the results.

| Our products feature innovative sustainable materials, _such as_ … | _According to the text_, Kai Fusser believes that traditional cardio … | 1. **Deliver Exceptional Service**: The foundation of word-of-mouth marketing is consistent excellence. Providing top-notch services or … | 1. Deliver Exceptional Service
• Consistently exceed customer expectations
• Focus on quality and attention … |
| _Certainly!_ If you're looking for cheese alternatives to replace Brie in your … | _Based on_ the text provided, _here_ are the key details about Armon Binns' … | **Ingredients:**
• 2 (3 oz) packages of orange-flavored Jello
• 1 cup tonic water (this is what … | Ingredients:
• 2 boxes orange-flavored Jello
• 1 can evaporated milk
• Tonic water … |
| _Overall_, while there are challenges, Tanzania is making progress … | _While_ many winter sports in the Pyrenees are similar to those found … | | |
| _Sure!_ Here's a simple guide to cooking a juicy salmon fillet: … | This _appears to_ be a fragment of poetry that creates a pastoral … | | |

ChatGPT      Claude      ChatGPT      Claude

(a) characteristic phrases      (b) unique markdown formatting

*Figure 2.* **Example responses from ChatGPT and Claude, showcasing their idiosyncrasies**: characteristic phrases (*left*) and unique markdown formatting (*right*). For clarity, we highlight each characteristic phrase with underline and model-specific color.

Across three groups of LLMs, the classification task benefits from seeing an increased number of tokens. Intriguingly, for chat APIs and instruct LLMs, we observe around 50% accuracy using only a single text token. This suggests that the initial token in a response contains certain distinctive signals for the classification problem. In Section 3.1, we provide further evidence supporting this observation.

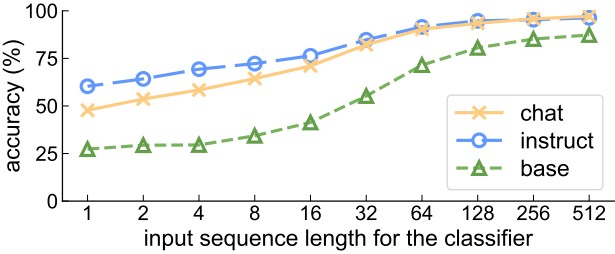

*Figure 3.* **Ablations on input length of text embedding models.** Classification accuracies improve as the text embedding models capture more context. Performance begins to saturate beyond an input sequence length of 256. Note that the three lines represent different groups of LLMs and are not directly comparable.

**Text embedding models.** We vary the underlying pretrained embedding models for sequence classification. The default setting we used in previous parts is fine-tuning the LLM2vec embedding models. We consider various generations of embeddings models spanning across architectures and training methods: ELMo (Peters et al., 2018), BERT (Devlin et al., 2018), T5 (Raffel et al., 2020), GPT-2 (Radford et al., 2019), and LLM2vec (BehnamGhader et al., 2024). Details on the fine-tuning setting can be found in Appendix A.2.

| method | chat | instruct | base |
|---|---|---|---|
| ELMo | 90.8 | 91.0 | 69.8 |
| BERT | 91.1 | 91.5 | 66.0 |
| T5 | 90.5 | 89.8 | 67.9 |
| GPT-2 | 92.1 | 92.3 | 80.2 |
| LLM2vec | **97.1** | **96.3** | **87.3** |

*Table 5.* **Different sequence embedding models.** LLM2vec achieves the best performance in classifying outputs from various LLMs among the five embedding models we study.

Table 5 shows the results. All sequence embedding models can achieve very high accuracies. The classification performance improves with more advanced sequence embedding models. Among all methods, LLM2vec demonstrates the best performance, achieving 97.1% on chat APIs, 96.3% on instruct LLMs, and 87.3% on base LLMs.

## 3. Concrete Idiosyncrasies in LLMs

In this section, we identify concrete idiosyncrasies in LLMs across three dimensions: words and letters, markdown formatting elements, and semantic meaning. For each dimension, we apply text transformations to isolate potential idiosyncrasies and assess their impacts on classification performance. We then highlight specific patterns within each dimension that distinguish LLMs.

### 3.1. Words and Letters

**Text shuffling.** To decouple the effects of words and letters from other factors, we remove special characters in LLM-generated responses, such as punctuations, markdown elements, and excessive white spaces. This ensures that each response consists solely of words separated by a white space. Additionally, we apply two shuffling strategies to the preprocessed text: word-level and letter-level shuffling. These transformations disrupt the natural order and force the classifier to learn patterns from raw text statistics. Table 6 presents the classification results.

| | chat | instruct | base |
|---|---|---|---|
| original | 97.1 | 96.3 | 87.3 |
| removing special characters | 95.1 | 93.8 | 75.4 |
| shuffling words | 88.9 | 88.9 | 68.3 |
| shuffling letters | 39.1 | 38.6 | 38.9 |

*Table 6.* **Classifications with only words and letters.** While removing special characters and shuffling words have little impact on accuracies, shuffling letters greatly reduces the performance.

Classifiers trained on responses without special characters achieve accuracies close to those using the original responses, *i.e.*, 95.1% for chat APIs, 93.8% for instruct LLMs, and 75.4% for base LLMs. Likewise, using word-

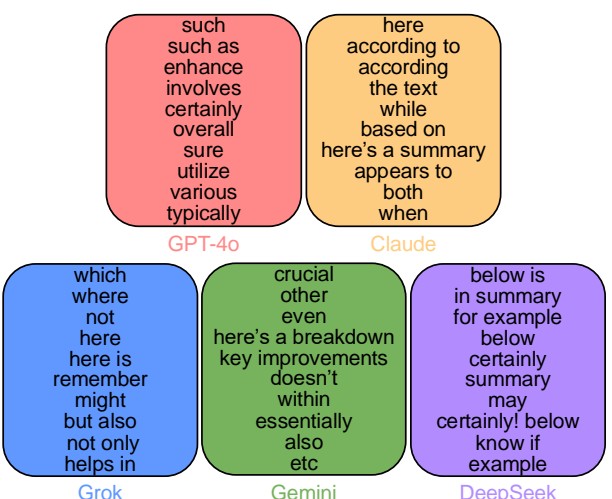

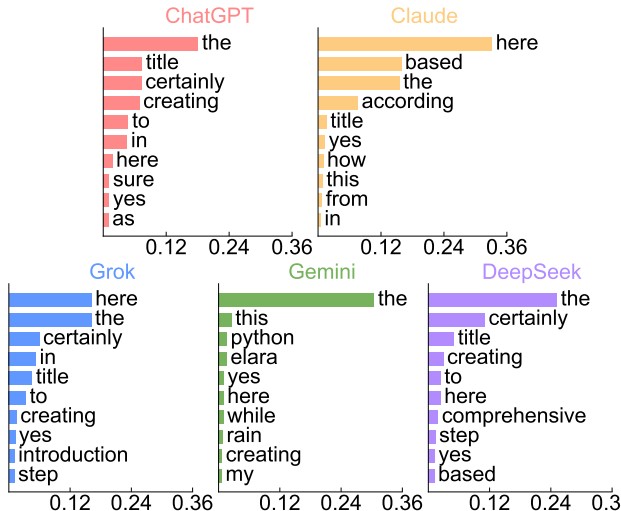

*Figure 4.* **Characteristic phrases.** We train a logistic regression model on TF-IDF features of chat APIs' outputs and extract the top 10 phrases for each LLM based on the coefficients of these features. We remove common words shared across these LLMs.

*Figure 6.* **First word.** We analyze the distribution of the first word in chat APIs' responses, with the top 10 most frequent words for each model. These differences in the first-word usage explain the non-trivial accuracy with only the first word in Figure 3.

shuffled responses yields high accuracies comparable to the original ones. Further, we plot the frequencies of several commonly used words from five chat APIs in Figure 5 (*left*). We observe distinct patterns among models, even for frequent English words: Claude has much lower frequencies for words like "the", "and", "to", and "of" than other chat APIs. These results suggest that *special characters and word order are not essential for distinguishing LLMs*; *word choices reflect substantial idiosyncrasies across models.*

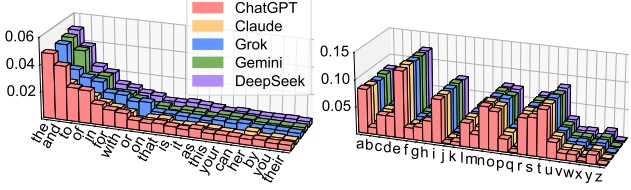

*Figure 5.* **Frequencies of words and letters.** The top 20 most frequently used words of LLMs (left) exhibit distinct patterns for each model, but their letter frequencies (right) are very similar. Results are on the chat API models.

In contrast, shuffling at the letter level results in a substantial drop in accuracy (49%-56%), approaching chance-level performance. This indicates that letter-level statistics alone are not sufficient for predicting LLM identities. To qualitatively visualize distinctions in letter distributions across models, Figure 5 (*right*) shows the frequency distribution of letters in responses generated by chat APIs. Different LLMs share almost identical letter distributions, indicating that *letters contribute minimally to idiosyncrasies in LLMs.*

**Characteristic phrases.** We use Term Frequency-Inverse Document Frequency (TF-IDF) to highlight characteristic

phrases inside LLM-generated responses that reflect each model's word choices. Formally, we treat each LLM response as a document and then extract TF-IDF features on all uni-gram and bi-gram words. We then train a $N$-way logistic regression model to predict the origin of responses on the extracted features. This simple linear classifier achieves 85.5% / 83.7% accuracy on chat APIs / instruct LLMs, close to 95.1% / 93.8% achieved with fine-tuning embedding models on responses without special characters (Table 6).

Since the coefficients of a logistic regression model provide a natural ranking for its features, we leverage these coefficients to highlight important phrases in the classification task. Figure 4 presents the top 10 phrases with the largest logistic regression coefficients for each of the five chat API models (excluding the 20 most frequently used words in Figure 4). Notably, these phrases often serve as transitions or emphasis in sentences. For example, ChatGPT likes to generate "such as", "certainly", and "overall", whereas Claude prefers "here", "according to", and "based on".

Figure 2a illustrates these characteristic phrases with example responses from ChatGPT and Claude. While ChatGPT begins responses with "certainly" and "below is", Claude usually references the original prompt using the phrases like "according to the text" and "based on the text". Moreover, Figure 6 reveals noticeable differences in the distribution of first word choices among chat APIs. Appendix B.5 provides characteristic phrases for other LLMs.

## 3.2. Markdown Formatting

We seek to understand how each LLM formats their responses, particularly in markdown. To this end, we focus on

common markdown elements used by LLMs: (1) bold text, (2) italic text, (3) header, (4) enumeration, (5) bullet point, (6) code block. We transform the LLM outputs by retaining only these formatting components while replacing other text with the marker "xxx". Appendix C provides examples of the transformed outputs. Table 7 shows the classification results after this transformation.

| | chat | instruct | base |
|---|---|---|---|
| original | 97.1 | 96.3 | 87.3 |
| markdown elements only | 73.1 | 77.7 | 38.5 |

*Table 7.* **Classifications with only markdown elements.** Using markdown elements can achieve high accuracies for chat APIs and instruct LLMs, but marginally better results for base LLMs.

Surprisingly, we observe our classifiers achieve high accuracies of 73.1% for chat APIs and 77.7% for instruct LLMs. However, the classification accuracies with base LLMs' responses are near chance-level guess (25%). This is likely because base LLMs tend to generate responses in plain text.

We count the occurrence of a markdown formatting element in each response. We then plot the distribution of these counts over all responses in Figure 7. Each model exhibits a unique way to format its responses. For instance, Claude (yellow) has a high density at zero in the bold text and header count distributions, indicating that it generates many responses without bold texts or headers. On the contrary, other LLMs exhibit lower values at zero and thus decorate text with these formatting elements more often.

Figure 2b visualizes how ChatGPT and Claude structure their responses in markdown. Interestingly, ChatGPT tends to emphasize each key point within enumerations in bold and highlight a title with markdown headers, but Claude formats text with simple enumeration and bullet points. More analysis for other models can be found in Appendix B.6.

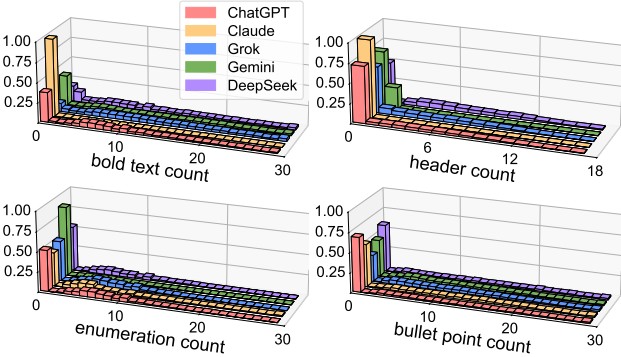

*Figure 7.* **Markdown formatting elements.** Each LLM has a distinctive distribution of markdown formatting elements.

## 3.3. Semantics

**Rewriting.** One potential reason for the high classification accuracy is the unique writing style (*e.g.*, word choice, sentence structure) of each LLM. To isolate this factor, we leverage another LLM (*e.g.*, GPT-4o mini) to rewrite LLM responses. Our rewriting approaches include (see Appendix C for example responses after rewriting):

* Paraphrasing: *Paraphrase the above text while maintaining the semantic meaning of the original text.*
* Translating: *Translate the above text into Chinese.*
* Summarizing: *Summarize the above text in one paragraph.*

| | chat | instruct | base |
|---|---|---|---|
| original | 97.8 | 96.3 | 87.3 |
| paraphrasing | 91.4 | 92.2 | 71.7 |
| translating | 91.8 | 92.7 | 74.0 |
| summarizing | 58.1 | 57.5 | 44.7 |

*Table 8.* **Classifications on rewritten responses.** Paraphrasing or translating LLM outputs achieves an accuracy comparable to that using original counterparts. However, summarizing these texts makes the model less capable of predicting LLM identities.

We show the results in Table 8 (see Appendix B.7 for results using alternative LLM for rewriting). The classifiers trained on paraphrased LLM responses maintain similar accuracy levels to those using original responses. Likewise, when using translated text, the classifiers are also able to differentiate between LLMs. These findings suggest that *the semantic meanings of words play a more significant role in predicting LLM origins than the exact word choice.*

Moreover, despite a noticeable accuracy drop (*i.e.*, >38%) with the summarized text, the resulting performance remains well above chance-level guess. This remarkable ability to classify the summarized texts shows the *high-level semantic difference in LLM-generated responses.*

**Open-ended language analysis.** In this part, we focus on studying the semantic difference in responses generated by LLMs. We employ another LLM (*e.g.*, ChatGPT) as a judge to provide open-ended, descriptive characterizations for each LLM's outputs. The results with other LLM judges for our language analysis are available in Appendix B.8.

Specifically, we present an LLM judge with two responses – generated by different models based on the same prompt – and ask it to analyze these responses from different angles (*e.g.*, tone and content). This process is repeated multiple times to gather a comprehensive collection of analyses. Finally, we query the LLM judge to summarize these analyses into bullet points that capture the characteristics of each model. The prompts are detailed in Appendix A.3.

The results of open-ended language analysis on ChatGPT *vs*. Claude are shown in Figure 8. For a detailed pairwise

| | |
|---|---|
| 1. **Descriptive and Detailed Tone:** Often uses narrative styles with an informative, engaging, or vivid tone. | 1. **Concise and Straightforward Tone:** Generally adopts a more direct, factual, or succinct tone. |
| 2. **Specific and Technical Word:** Employs descriptive and technical vocabulary, enhancing depth and specificity. | 2. **Functional and Clear Word Choices:** Prefers simple or action-oriented language prioritizing clarity and practicality. |
| 3. **Structured and Contextual Opening Lines:** Typically begins with context-setting or narrative introductions. | 3. **Immediate and Direct Opening Lines:** Often starts with a straightforward statement or summary without extended context. |
| 4. **Markdown Formatting for Organization:** Utilizes various markdown elements like headings, lists, and bold text for clarity. | 4. **Minimal Markdown or List Use:** Relies on plain lists or simple formatting for quick reference. |
| 5. **Comprehensive and In-Depth Content:** Offers rich detail, focusing on explanations, background, and broader topics. | 5. **Focused and Summarized Content:** Concentrates on essential points and specific phenomena, avoiding extensive detail. |
| ChatGPT | Claude |

*Figure 8.* **Results of our open-ended language analysis on ChatGPT and Claude. ChatGPT** features descriptive language, sophisticated markdown formatting, and in-depth details, while **Claude** highlights straightforward tone, minimal structure, and summarized content.

comparison of the responses, see Figure 17 in Appendix C. ChatGPT is characterized by descriptive and detailed responses in an engaging tone. In contrast, Claude prioritizes simplicity with only key points and straightforward language. Additional results on chat API models and instruct LLMs are provided in Appendix B.8.

## 4. Implications

In this section, we explore the broader implications of our framework, regarding synthetic data and model similarity.

**Idiosyncrasies via synthetic data.** Using synthetic data has become a common practice when training frontier LLMs (Abdin et al., 2024a;b; Liu et al., 2024). We conduct supervised fine-tuning (SFT) on two base LLMs (Llama3.1-8b and Gemma2-9b) using Ultrachat, *i.e.*, dialogues generated by ChatGPT. After the SFT stage, we train a classifier to distinguish between responses from two fine-tuned models. We find that SFT on the same synthetic dataset significantly reduces the classification accuracy from 96.5% to 59.8%, narrowing down the differences between these two models.

In addition, we generate responses from Llama3.1-8B and Gemma2-9B in instruct LLMs using UltraChat prompts. Then we fine-tune Qwen2.5-7B base LLM on each set of responses respectively. Interestingly, responses from the two resulting fine-tuned models can be classified with 98.9% accuracy, suggesting that each fine-tuned model retains the unique characteristics in its SFT data. These findings suggest that training with synthetic data can propagate the idiosyncrasies in the source model.

We note that this behavior is not limited to synthetic datasets; in fact, training on different datasets often leads to idiosyncratic patterns in model outputs (Mansour & Heckel, 2024).

**Inferring model similarity.** Our framework offers a quantitative approach for assessing similarities between proprietary and open-weight LLMs. Given a set of $N$ LLMs, we omit one model and train a classifier on responses from the remaining $N - 1$ models. We then evaluate which LLM the classifier associates the responses of the excluded model with. The model that is most frequently predicted as the

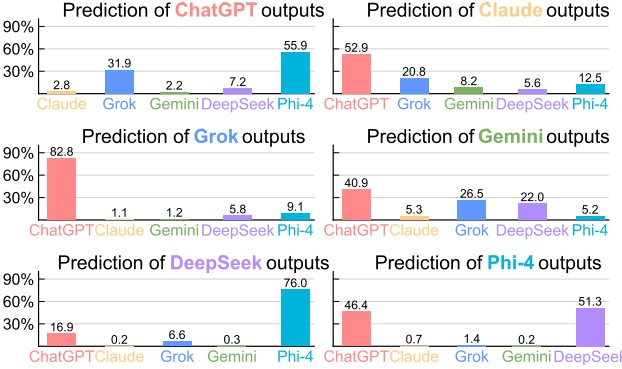

*Figure 9.* **Inferring model similarity.** We consider 6 LLMs, including 5 chat API models and Phi-4. In each subfigure, we evaluate a five-way classifier on outputs from the excluded LLM and present the distribution of predicted model origins. There is a strong tendency for LLM outputs to be predicted as ChatGPT.

source is considered the closest match to the excluded LLM. This process is repeated for each of the $N$ models. For this analysis, we include the open-weight Phi-4 (Abdin et al., 2024b) alongside 5 chat API models. Note that Phi-4 uses a substantial amount of synthetic data in its training.

Figure 9 shows the results. Intriguingly, for Claude, Grok, and Gemini, we observe a strong tendency for their outputs to be classified as ChatGPT. For instance, when Grok is the excluded model, 82.8% of its responses are classified as ChatGPT. In addition, responses from ChatGPT and DeepSeek are frequently identified as coming from Phi-4, with 55.9% and 76.0% of their responses, respectively. In turn, most of Phi-4's outputs are classified as originating from ChatGPT or DeepSeek.

**Robust evaluation of LLMs.** Our findings reveal a potential vulnerability in widely used LLM evaluation methodologies. It has become a common strategy to incorporate human judgement in evaluating LLMs, for instance, Chatbot Arena (Chiang et al., 2024). It is a voting-based leaderboard where users submit preferences of the responses from two randomly chosen models. This benchmark has gained significant traction and is now a key reference point for frontier

model development. However, exploiting the idiosyncratic property of LLM outputs, a malicious attacker can identify the model behind the candidate responses and consistently vote for the target model, thereby manipulating the leaderboard rankings. Concurrent work by Huang et al. (2025) has demonstrated the feasibility of this attack in simulation. We hope our work brings attention to potential weaknesses in current evaluation pipelines, as they can misguide model development and optimization efforts (Singh et al., 2025).

## 5. Related Work

**Dataset classification.** Torralba & Efros (2011) introduced the "Name That Dataset" experiment a decade ago to highlight the bias present in visual datasets at that time. Recently, Liu & He (2025) revisited this problem (termed dataset classification) and found that current large-scale, supposedly more diverse visual datasets are still very biased. Zeng et al. (2024) further identified structural and semantic components in images as key contributors to these biases. You et al. (2024) and Mansour & Heckel (2024) applied the dataset classification framework to study bias in synthetic images and LLM pretraining datasets, respectively. While the synthetic task shown in Figure 1 is conceptually similar to dataset classification, we focus not on training datasets but on the distinctive characteristics inherent to LLMs. Shortly after our initial arXiv preprint, Suzuki et al. (2025) demonstrated that subtle variations in the training process can result in distinguishable outputs across LLMs.

**Human *vs*. machine-generated texts.** Many prior works have studied the problem of determining whether a text is authored by a human or an AI system (Mitchell et al., 2023; Wu et al., 2023; Cai et al., 2025). Model-free approaches typically use linguistic properties such as n-gram frequencies (Badaskar et al., 2008; OpenAI, 2019), entropy (Lavergne et al., 2008; Gehrmann et al., 2019) or negative probability curvature (Mitchell et al., 2023; Bao et al., 2024). Other works leverage neural network features to perform this task, such as fine-tuning BERT (Uchendu et al., 2021; Ippolito et al., 2020). Russell et al. (2025) found that experienced chatbot users are good at distinguishing between AI- and human-written articles. Neural authorship attribution (Uchendu et al., 2020; Antoun et al., 2023; Huang et al., 2024) seeks not only to identify machine-generated text but also to attribute it to specific text generators. In this work, we focus on the distinguishability between LLMs rather than between AI *vs*. human.

**Characteristics of machine-generated texts.** Beyond detection, it is also important to understand the distinctive properties of machine-generated texts. N-gram frequencies have long served as a basic signal for such purpose. McGovern et al. (2024) showed that LLM outputs contain unique lexical and syntactic fingerprints that distinguish them from human writing. Other works have examined stylometric features in AI-generated content (Zaitsu & Jin, 2023). Most recently, Chakrabarty et al. (2024) studied the human-AI alignment in the writing process, finding that professional writers can effectively identify and edit undesirable idiosyncrasies common in LLM-generated text. Our work seeks to provide a deeper and principled understanding behind the observed idiosyncrasies from our framework in Figure 1.

**Understanding differences between distributions.** A line of research (Dunlap et al., 2024; Zhong et al., 2024) has used foundation models to describe qualitative differences between pairs of data distributions (*e.g.*, image datasets). Gao et al. (2025) and Cai et al. (2025) conducted auditing studies on LLM APIs to detect instances of model substitution, such as watermarking and quantization. The most relevant work to us is Dunlap et al. (2025), which proposed VibeCheck to understand user-aligned traits in LLM outputs. They found that LLMs often vary in styles, such as being more formal or friendly. In contrast, our work aims to identify broader generalizable patterns to interpret the high classification performance.

## 6. Conclusion

We demonstrate the presence of idiosyncrasies in Large Language Models (LLMs) and investigate a synthetic task designed to quantify their extent. We find that simply fine-tuning pretrained text embedding models on LLM outputs leads to exceedingly high accuracy in predicting the origins of the text. This phenomenon persists across diverse prompt datasets, LLM combinations, and many other settings. We also pinpoint concrete forms of these idiosyncrasies within LLMs. We hope our work encourages further research into understanding idiosyncrasies in LLMs.

We conclude by outlining several directions for future work:

- It is worth investigating whether our observations generalize to LLMs beyond the Transformer architecture, such as state space models (Gu & Dao, 2024) and diffusion-based language models (Nie et al., 2025).

- Understanding how the training process results in these idiosyncrasies remains an important open question.

- Third, our setup does not consider scenarios where the list of source LLMs could be large and even unknown beforehand, which require further investigation. Our initial results suggest it is promising: a 10-way classification problems involving 5 chat APIs and 5 instruct LLMs including Phi-4 achieves 92.2% accuracy.

- It is interesting to examine how these idiosyncrasies relate to model distillation (Hinton et al., 2015), a technique that has become increasingly prevalent in practice.

## Acknowledgments

We thank Zekai Wang for valuable discussions. Mingjie Sun was supported by funding from the Bosch Center for Artificial Intelligence.

## Impact Statement

Our study investigates the distinguishability of LLMs. On the positive side, our results offer insights into LLM behaviors that enhance transparency, accountability and provenance in tracking AI generated content. This has valuable applications such as combating misinformation, and ensuring responsible use of LLMs. However, our results highlight challenges in model training. Reliance on synthetic data or model distillation risks propagating biases and potentially infringing on the intellectual property of proprietary models.

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

# Appendix

## A. Implementation Details

### A.1. Response Generation

We report our procedure for generating responses from chat APIs, instruct LLMs, and base LLMs. For chat APIs, we access a stable version of each model, including GPT-4o-2024-08-06, Claude-3.5-Sonnet-20241022, Grok-Beta, Gemini-1.5-Pro-002, and DeepSeek-Chat, through its official API between November 28, 2024, and February 6, 2025, generating responses with their default sampling setting. For instruct LLMs, we use greedy decoding to sample outputs. For base LLMs, we set the temperature to $T = 0.6$ and apply a repetition penalty of 1.1 to avoid repetitive completions.

### A.2. Training Setup

In this part, we describe our fine-tuning process using the text embedding models on LLM responses. We use the first 512 tokens of each generated response for training and evaluation. To perform sequence classification, we add a linear layer as the classification head on top of each text embedding model. For ELMo, BERT, LLM2vec, this layer is applied to the average embeddings over all tokens in a sequence. For T5 and GPT-2, we follow the original setups (Radford et al., 2019; Raffel et al., 2020) and apply the head on the output of the last token.

For smaller text embedding models, such as ELMo, BERT, T5, and GPT-2, we fine-tune the entire model along with the classification head, searching over base learning rates {3e-3, 1e-3, 3e-4, 1e-4, 3e-5, 1e-5, 3e-6, 1e-6}. For the largest LLM2vec model, we employ the parameter-efficient LoRA (Hu et al., 2022) fine-tuning method with a rank of 16, LoRA $\alpha$ of 32, a dropout rate of 0.05, and a base learning rate of 5e-5. Table 9 details our basic training recipe.

| config | value |
|---|---|
| optimizer | AdamW |
| weight decay | 0.001 |
| optimizer momentum | $\beta_1, \beta_2 = 0.9, 0.999$ |
| training epochs | 3 |
| batch size | 8 |
| learning rate schedule | cosine decay |
| warmup schedule | linear |
| warmup ratio | 10% |
| gradient clip | 0.3 |

*Table 9.* Our fine-tuning recipe.

### A.3. Prompts for Open-ended Language Analysis

We detail the procedures of our open-ended language analysis in Section 3.3. Given the same input, we sample a pair of responses from two LLMs and present them, along with an analysis prompt (see Figure 10a), to an LLM judge for comparison. To avoid the LLM judge exploiting any prior knowledge of the models, we anonymize model identities using an index distribution. This process is repeated for 35 response pairs, yielding a set of detailed analyses. Finally, we use the summarization prompt (see Figure 10b) to distill these analyses into 5 bullet points that characterize the idiosyncrasies of each model.

> Here are some samples from 2 different distributions of text.
>
> Sample from distribution {1, 2}: {text sample 1}
> Sample from distribution {1, 2}: {text sample 2}
>
> Determine the unique characteristics of the 2 distributions, and summarize them with five bullet points for each. Each bullet point should analyze the response from the same angle and can be easily compared by a human. Focus on key aspects that differentiate the distributions. Focus on the overall tone, specific word choices, opening lines, markdown formatting, and content. Output only 5 bullet points per distribution, no additional text.

(a) analysis prompt

> Below are 35 summaries that compare the unique features in 2 text distributions point by point.
>
> {Analysis 1}
> {Analysis 2}
> ...
> {Analysis 35}
>
> Condense all 35 summaries into a single summary. Focus on how each summary evaluates the overall tone, specific word choices, opening lines, markdown formatting, and content. Provide exactly five bullet points for each distribution, one for each feature. Each bullet point should consist of a descriptive title characterizing the feature and a short sentence explaining it concisely. Emphasize recurring and significant features, remove the redundant ones, and ensure the titles and sentences clearly differentiate the distributions for easy human comparison. Output only 5 bullet points per distribution, no additional text.

(b) summarization prompt

*Figure 10.* Prompts in our open-ended language analysis.

# B. Additional Results

### B.1. Controlled Studies

We present additional studies on our classification framework, to complement our results in Section 2.2.

**Sampling methods.** We consider outputs when sampled using different decoding strategies. Specifically, we use four widely used sampling methods: greedy decoding, temperature softmax, top-k, and top-p sampling. For each method, we generate a set of responses from the same LLM. We then fine-tune the LLM2vec embedding model to predict the sampling method responsible for each response.

|         | greedy | softmax | top-k | top-p |
|---------|--------|---------|-------|-------|
| greedy  | -      | -       | -     | -     |
| softmax | 59.6   | -       | -     | -     |
| top-k   | 58.2   | 50.0    | -     | -     |
| top-p   | 52.9   | 51.0    | 52.1  | -     |

*Table 10.* **Classifications with different sampling methods.** Distinguishing responses generated by the same model using different sampling strategies is only marginally better than chance accuracy. The results are on Llama3.1-8b instruct model's responses.

Table 10 presents the results for all pairs of sampling methods. Notably, the accuracy of distinguishing between responses generated by the same LLM remains relatively low, with the highest accuracy across all configurations being 59%. Furthermore, in a more fine-grained 5-way classification task distinguishing softmax sampling at five different temperatures ($T = 0, 0.25, 0.5, 0.75, 1$), we obtain an accuracy of 37.9%, only marginally better than the random chance level of 20%. These results suggest that outputs from the same LLM are not easily separable based on decoding strategies.

**Training data size.** We vary the number of training samples generated by LLMs and train the classifier with the same total number of iterations. We present the results in Figure 11. The classification performance increases with more training samples. This trend holds consistently across chat APIs, instruct LLMs, and base LLMs. Furthermore, as few as 10 samples, the classifier achieves non-trivial accuracy (*e.g.*, 40.3% on chat APIs), surpassing 20% chance-level guess.

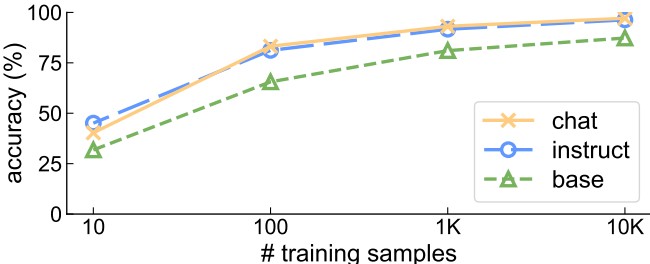

*Figure 11.* **Different numbers of training samples.** Our sequence classifiers benefit from more training samples. The classification performance converges when using about 10K training samples.

## B.2. Text Similarities

Here we use text similarity metrics to quantify differences between LLM outputs. We consider three standard metrics – ROUGE-1 (Lin, 2004), ROUGE-L (Lin, 2004), and BERTScore (Devlin et al., 2018) – to measure lexical and semantic similarity. We compute the mean F1-score for each metric across all response pairs generated by any two different chat API models given the same prompt. For comparison, we also measure the similarity between responses sampled within the same model. As shown in Table 11, responses from different LLMs exhibit lower text similarities than those from the same model.

|  | across LLMs | within an LLM |
|---|---|---|
| ROUGE-1 | 0.499 | 0.660 |
| ROUGE-L | 0.256 | 0.414 |
| BERTScore* | 0.220 | 0.482 |

*Table 11.* **Text similarity scores.** We evaluate the text similarity of LLM outputs using ROUGE-1, ROUGE-L, and BERTScore. Responses from different LLMs exhibit low lexical similarity.

## B.3. Confusion Matrix

In Figure 12, we present the confusion matrix for the $N$-way classifiers that are trained on responses generated by chat APIs, instruct LLMs, and base LLMs, respectively. The results demonstrate that our classifiers can accurately predict the origin of LLM-generated responses, with minimal confusion between different LLMs.

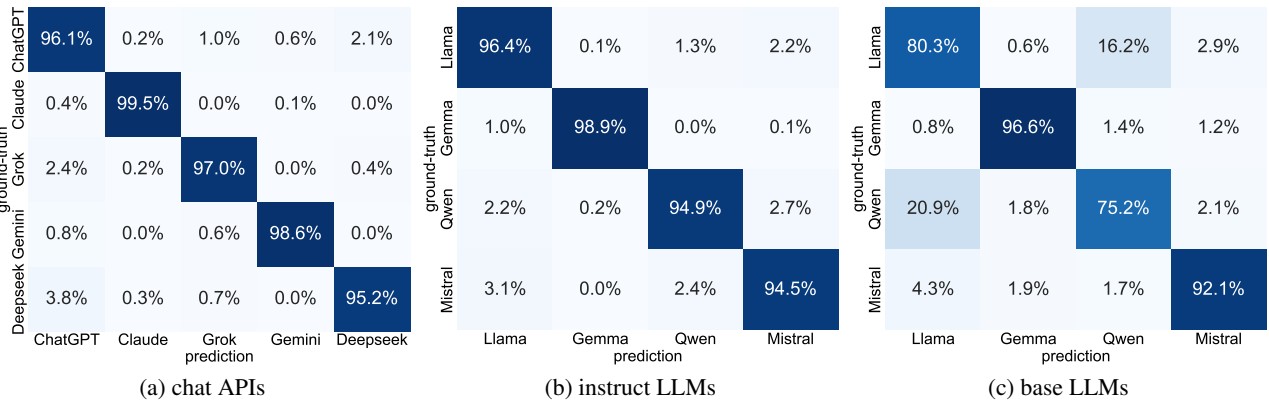

(a) chat APIs        (b) instruct LLMs        (c) base LLMs

*Figure 12.* Confusion matrices for $N$-way classifiers on three groups of LLMs: chat APIs, instruct LLMs, and base LLMs.

## B.4. Words and Letters

Figure 13 presents the frequencies of the 20 most commonly used words (*left*) and all English letters (*right*) across instruct and base LLMs. Consistent with our observations in Section 3.1, we find notable differences in the distribution of commonly used words between these models, such as "the", "and", "to". In contrast, the letter distributions are nearly identical.

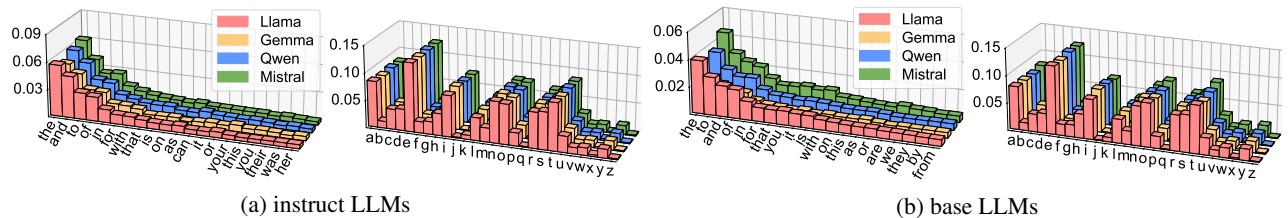

(a) instruct LLMs  (b) base LLMs

Figure 13. Word and letter frequencies in instruct and base LLMs.

## B.5. Characteristic Phrases

We provide additional results for characteristic phrases as presented in Section 3.1. We follow the same methodology in Figure 4 to extract characteristic phrases of instruct and base LLMs. Specifically, we train a four-way logistic regression classifier on the TF-IDF features of their responses and use the coefficients to select important phrases of each model.

As shown in Figure 14, each instruct LLM contains quite distinct characteristic phrases. For example, Llama frequently employs terms "including" and "such as" to introduce specific examples in the output, whereas Gemma tends to engage with users using phrases "let me" and "know if". In contrast, the extracted phrases from base LLMs are less distinctive, primarily consisting of common words such as "the", "to", and "you".

Figure 15 illustrates the distribution of first word choices in instruct and base LLMs. Similar to chat APIs (Figure 6), instruct LLMs display varied distributions. However, base LLMs exhibit substantial overlap in their most frequent first words, *e.g.*, "the", "and", "of", "to", and "in".

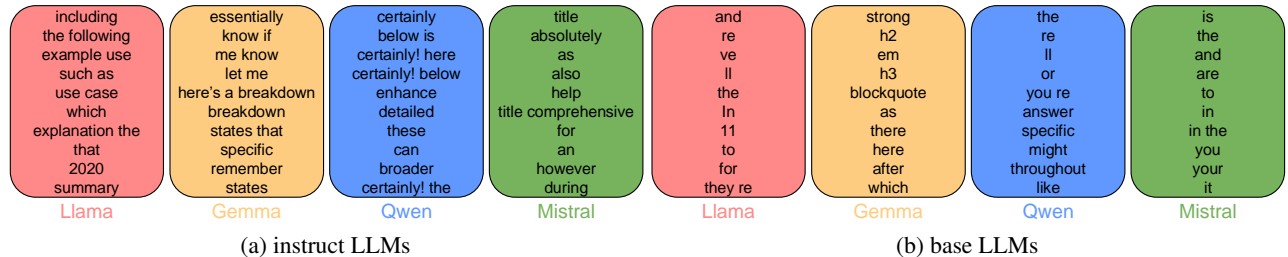

(a) instruct LLMs  (b) base LLMs

Figure 14. Characteristic phrases for instruct[2] and base LLMs.

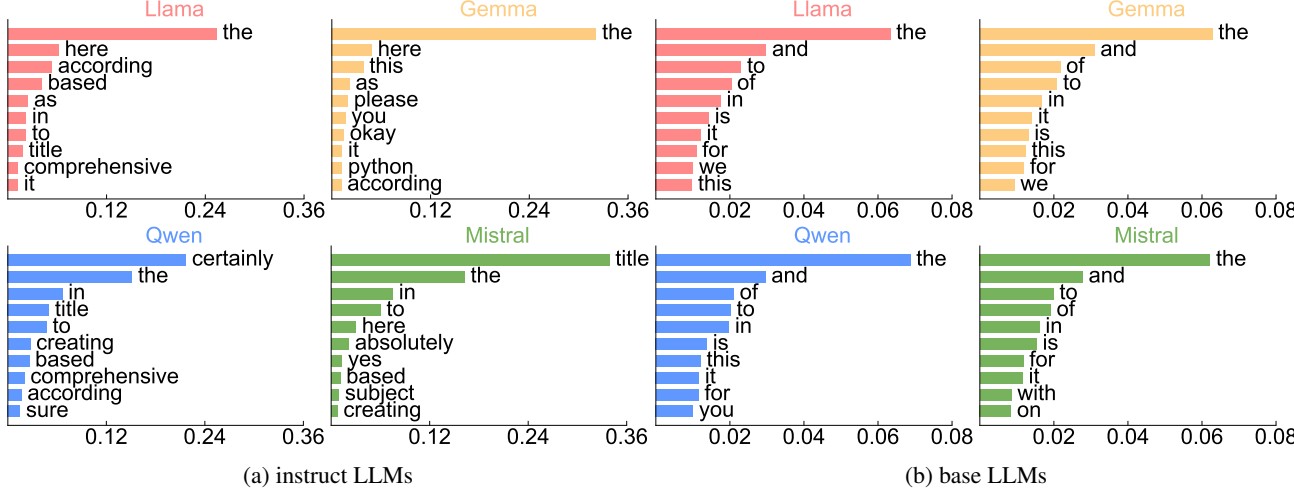

(a) instruct LLMs  (b) base LLMs

Figure 15. Distribution of first word choices in instruct and base LLMs.

---

[2]In Llama of instruct LLMs, the phrase "explanation the" corresponds to a markdown header or bold text for "explanation" followed by a new sentence starting with "the".

## B.6. Unique Markdown Formatting

In this part, we provide additional results for the analysis of markdown formatting as presented in Section 3.2. Figure 16 illustrates the distribution counts of six markdown formatting elements across different models. For both chat API models (Figure 16a) and instruct LLMs (Figure 16b), we observe distinct differences in the usage of bold texts, headers, enumerations, and bullet points, while italic texts show less variation. Intriguingly, Gemini uses much more italic texts (a lower density at zero in the italic text) than other chat APIs, where similar observations can be found on Gemma2.

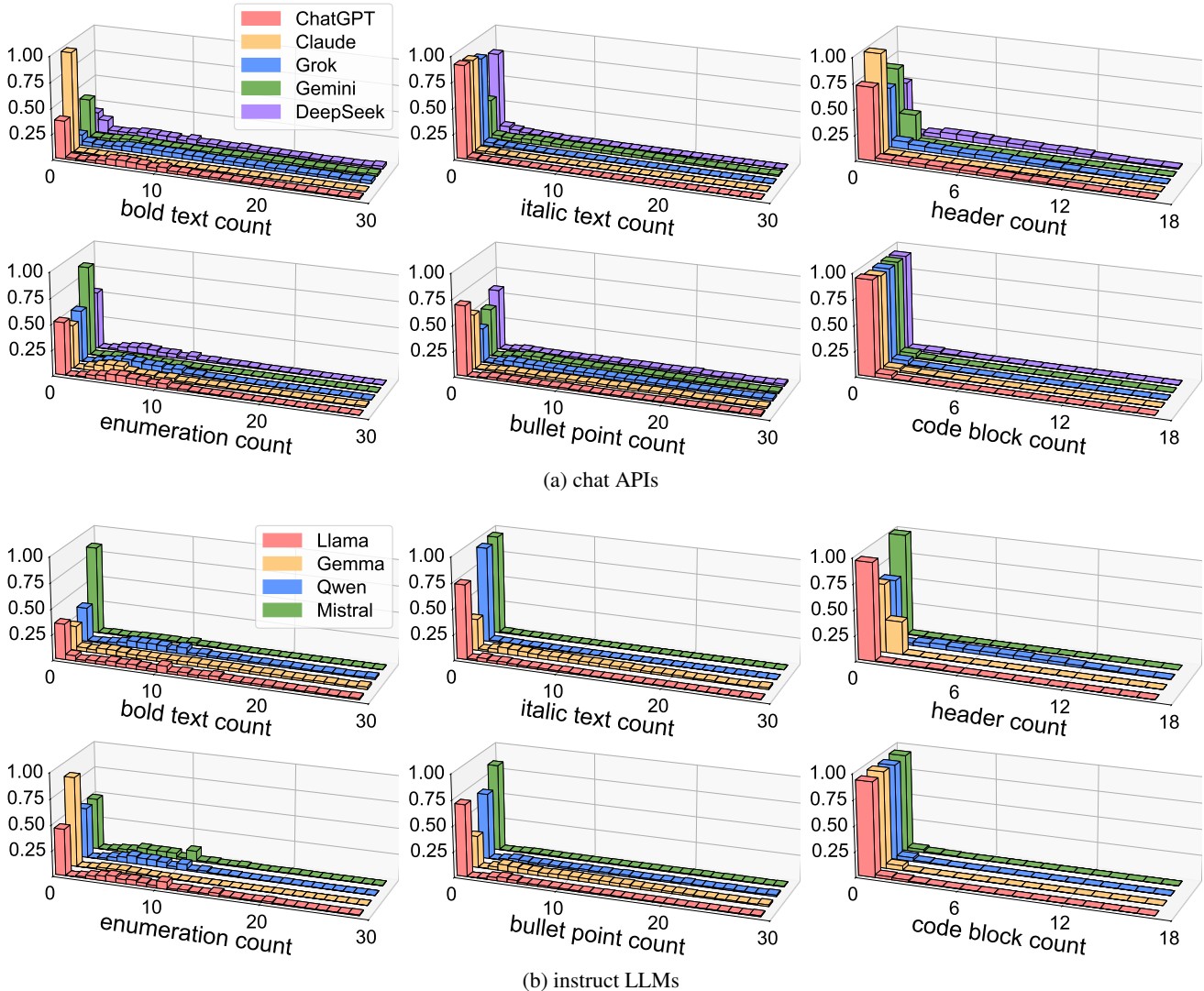

*Figure 16.* Markdown formatting elements for chat APIs (*top*) and instruct LLMs (*bottom*).

## B.7. Rewriting LLM outputs

In Section 3.3, we used GPT-4o-mini to rewrite LLM outputs. Here, we present results using an alternative model: Qwen2.5-7B-Chat. As shown in Table 12, our observations remain consistent, indicating that our findings are robust to the choice of LLM used for rewriting.

| LLM for rewriting | original | paraphrase | translate | summarize |
|---|---|---|---|---|
| GPT-4o-mini | 97.8 | 93.6 | 93.9 | 63.7 |
| Qwen2.5-7B-Instruct | 97.8 | 92.6 | 94.3 | 71.5 |

*Table 12.* **Classification accuracies on rewritten responses.** The results are on Chat API responses.

## B.8. Open-ended Language Analysis

**Ablation on LLM judges.** Here we demonstrate our findings in Figure 8 of Section 3.3 remains consistent under several LLM judges. Specifically, we change the LLM judge from ChatGPT to Claude, Grok, and Gemini. We show the results in Figure 17. Regardless of the choice of LLM judges, our language analysis reveals that ChatGPT often uses detailed explanations and complex formatting structures, whereas Claude emphasizes key contents without extensive elaboration.

• Tone: Authoritative and formal, using academic language with detailed explanations and complex sentence structures
• Word Choice: Rich descriptive vocabulary with technical terms, metaphors, and evaluative language ("renowned," "instrumental," "exquisite")
• Opening Style: Direct statements or enthusiastic "Certainly!" responses that establish context and preview content
• Markdown Usage: Extensive formatting with bold headers, nested bullet points, and hierarchical section markers
• Content Structure: Comprehensive paragraphs with detailed explanations, examples, and flowing transitions between ideas

ChatGPT

• Tone: Analytical and cautious, frequently referencing source material and acknowledging limitations
• Word Choice: Simpler, more practical vocabulary focused on clarity and accessibility rather than style
• Opening Style: Meta-references like "According to the text" or explicit disclaimers about content limitations
• Markdown Usage: Minimal formatting with basic bullet points or simple numbered lists
• Content Structure: Concise bullet points and brief statements organized in categorical lists without extensive elaboration

Claude

(a) Claude as the LLM judge.

- **Tone: Warm and Informative** - Uses an inviting, educational tone to engage and inform the reader.
- **Word Choices: Descriptive and Detailed** - Employs rich, technical, or emotive language to enhance understanding or evoke emotion.
- **Opening Lines: Contextual and Engaging** - Provides context or a narrative hook to draw readers in.
- **Markdown Formatting: Structured and Emphasized** - Extensive use of headings, lists, and bold text for clarity and organization.
- **Content: Comprehensive and Insightful** - Focuses on in-depth explanations, examples, and broad implications.

ChatGPT

- **Tone: Direct and Practical** - Often concise, focusing on actionable insights or straightforward information.
- **Word Choices: Simple and Functional** - Prefers straightforward language, emphasizing utility over emotional or detailed descriptions.
- **Opening Lines: Immediate and Focused** - Jumps directly into the topic or lists key points without much preamble.
- **Markdown Formatting: Minimalist** - Uses basic markdown elements like bullet points or simple lists for clarity.
- **Content: Succinct and Focused** - Provides essential information with less depth, often aiming for quick readability or practicality.

Claude

(b) Grok as the LLM judge.

• **Formal Tone:** Often adopts a formal, academic, or instructional tone, prioritizing clarity and detail.
• **Precise Diction:** Employs specific, descriptive, and sometimes technical vocabulary relevant to the subject matter.
• **Contextual Openings:** Typically begins with introductory statements, context-setting, or clear purpose statements.
• **Structured Formatting:** Frequently utilizes headings, lists, bolding, and other markdown features for organization and emphasis.
• **Comprehensive Content:** Provides detailed explanations, examples, and in-depth information, often covering a broader scope.

ChatGPT

• **Informal Tone:** Generally uses a more informal, conversational, or concise tone, prioritizing brevity and accessibility.
• **Simplified Language:** Favors simpler, more direct vocabulary, avoiding jargon and focusing on key information.
• **Direct Openings:** Often starts directly with the main point or a summary of key information.
• **Minimal Formatting:** Uses less markdown formatting, relying primarily on basic lists, bullet points, or plain text.
• **Concise Content:** Focuses on summarizing key points, offering practical advice, or providing concise overviews.

Claude

(c) Gemini as the LLM judge.

*Figure 17.* Results of our open-ended language analysis on ChatGPT and Claude with different LLM judges.

**Open-ended language analysis results on other LLMs.** In Section 3.3, we presented the results of open-ended language analysis for ChatGPT and Claude. Here, we extend our analysis to other chat API models and instruct LLMs. The full results are shown in Figure 18, where we use ChatGPT as the LLM judge to compare responses generated by two models within the same category (chat APIs / instruct LLMs). Our analysis highlights several interesting characteristics of each model. For example, Grok's responses tend to feature rich language and comprehensive content, whereas Gemini's outputs are more concise with direct openings.

- **Descriptive Tone:** Emphasizes detail, narrative, or objective perspectives, often exploring themes with depth and nuance.
- **Rich Language:** Incorporates emotive, analytical, or descriptive language to enhance understanding and engagement.
- **Detailed Openings:** Introduces the subject with context, thematic elements, or guiding points, establishing a comprehensive view.
- **Structured Formatting:** Utilizes formatting elements like headings, lists, and bold text to organize information clearly.
- **Comprehensive Content:** Delivers in-depth analysis or exploration of subjects, highlighting diverse factors or broader implications.

Grok

- **Concise Tone:** Often adopts a straightforward, direct, or expressive approach focusing on succinctness and dramatic tension.
- **Specific Language:** Utilizes precise or vivid language to convey themes like tension, urgency, and significant events or outcomes.
- **Direct Openings:** Typically begins with immediate context or action, setting a focused tone or premise from the start.
- **Simple Formatting:** Relies mainly on plain text or minimal formatting to maintain clarity and directness.
- **Core Content:** Centers on key insights or motivations, providing essential information or narratives with limited embellishment.

Gemini

(a) chat APIs

- **Tone Characterization:** Often formal, factual, and exacting, emphasizing detailed analysis and structured presentations.
- **Word Choices Depiction:** Technical and precise, leveraging industry-specific or academic terminology to convey depth and credibility.
- **Opening Line Initiation:** Frequently starts directly with context, setting an immediate formal or informative tone.
- **Markdown Use:** Prefers structured formats with headings, bullet points, or numbered lists to enhance clarity.
- **Content Focus:** Delivers comprehensive, detailed content with substantial data, examples, or specific tactics.

Llama

- **Tone Characterization:** Generally more conversational, engaging, and accessible, often blending storytelling or personal insights.
- **Word Choices Depiction:** Simple, expressive, and often emotive language, creating relatability and narrative flair.
- **Opening Line Initiation:** Begins engagingly, often with context or an emotional hook, to capture the reader's interest.
- **Markdown Use:** Employs varied markdown elements like bullet points or emphasis for readability and structure.
- **Content Focus:** Centers on overarching concepts, insights, and experiences, often prioritizing engagement over exhaustive detail.

Gemma

- **Detailed and Analytical Tone:** The tone is often formal, focusing on analysis, exploration, and organized narratives.
- **Descriptive and Formal Word Choices:** Uses technical and descriptive vocabulary that highlights complexity and depth.
- **Comprehensive Opening Lines:** Starts often with broad context or detailed introductions, setting a formal framework.
- **Structured Markdown Formatting:** Utilizes headers, sections, and lists to enhance readability and organization.
- **Content Depth and Exploration:** Provides comprehensive narratives with detailed discussions, focusing on systematic approaches and analysis.

Qwen

- **Concise and Direct Tone:** The tone is generally straightforward, practical, and sometimes conversational, focusing on direct explanations.
- **Simplified and Emotive Word Choices:** Prefers straightforward language, occasionally prioritizing emotional or motivational impact.
- **Immediate Opening Lines:** Begins directly with subject matter or actions, often in conversational or narrative styles.
- **Minimalist Markdown Formatting:** Uses simpler formatting with lists or plain text to maintain clarity without embellishments.
- **Summary Focused Content:** Concentrates on key points with succinct explanations, prioritizing clarity and accessibility over comprehensive detail.

Mistral

(b) instruct LLMs

*Figure 18.* Additional results of our open-ended language analysis on chat APIs (*top*) and instruct LLMs (*bottom*).

# C. Response Demonstrations

In this part of the appendix, we present examples of LLM responses. Table 13 and 14 illustrate responses before and after our prompt-level interventions (Section 2.2). Table 15 shows the transformed responses when only markdown elements are retained (Section 3.2).

Table 16 presents the LLM rewritten responses (Section 3.3). For paraphrasing, we find that formatting style remains largely unchanged, *e.g.*, the number of enumerated lists are the same. Most of the differences lie in their word choices, paraphrased texts use different words with similar meanings but do not change the high-level semantic meaning of the original texts.

Additionally, Table 17 provides a pairwise comparison of responses, supporting our findings in Table 8. Table 18, 19, 20, 21, 22, 23, 24, 25, 26, 27, 28, 29, 30 contain example responses from each LLM considered in Section 2, including ChatGPT, Claude, Grok, Gemini, DeepSeek, Llama3.1-8b (instruct), Gemma2-9b (instruct), Qwen2.5-7b (instruct), Mistral-v3-7b (instruct), Llama3.1-8b (base), Gemma2-9b (base), Qwen2.5-7b (base), and Mistral-v3-7b (base).

---

Can you suggest any other cheese alternatives that I can use instead of Brie for this recipe? Generate according to: Add one or two slices Brie o the beef and let it melt. Cut the green apple in thin slices. Top with the mixed salad and finish with the apple slices.

---

If you're looking for cheese alternatives to Brie for your recipe, consider the following options:
1. **Camembert**: This cheese is quite similar to Brie in texture and flavor, offering a creamy consistency and a mild, earthy taste that will complement the beef and apple.
2. **Taleggio**: An Italian cheese with a soft texture and a tangy, fruity flavor. It melts well and can add a unique twist to your dish.
3. **Fontina**: Known for its smooth melting properties, Fontina has a nutty and slightly sweet flavor that pairs nicely with beef and apples.
4. **Goat Cheese**: If you prefer a tangier option, goat cheese can provide a creamy texture and a distinct flavor that contrasts well with the sweetness of the apple.
5. **Havarti**: A semi-soft cheese with a buttery and slightly sweet taste, Havarti melts beautifully and can be a great substitute for Brie.

These alternatives should work well in your recipe, providing a delicious and melty addition to your beef and apple combination.

*(a) original*

---

Can you suggest any other cheese alternatives that I can use instead of Brie for this recipe? Generate according to: Add one or two slices Brie o the beef and let it melt. Cut the green apple in thin slices. Top with the mixed salad and finish with the apple slices.

Please provide a concise response in a single paragraph, limited to a maximum of 100 words.

---

For a delicious alternative to Brie, consider using Camembert or triple cream cheese, both of which offer a similar creamy texture and rich flavor. If you prefer a slightly tangier option, goat cheese can provide a delightful contrast to the beef and apple. For a milder taste, Havarti or Fontina can melt beautifully and complement the dish's flavors. Each of these cheeses will add a unique twist while maintaining the creamy, melt-in-your-mouth experience that Brie provides.

*(b) length control*

---

Can you suggest any other cheese alternatives that I can use instead of Brie for this recipe? Generate according to: Add one or two slices Brie o the beef and let it melt. Cut the green apple in thin slices. Top with the mixed salad and finish with the apple slices.

Please provide your response in plain text only, avoiding the use of italicized or bold text, lists, markdown, or HTML formatting.

---

You can use Camembert or triple cream cheese as alternatives to Brie in your recipe. Both have a similar creamy texture and rich flavor that will complement the beef and other ingredients.

*(c) format control*

*Table 13.* Examples of prompt-level interventions on ChatGPT.

Can you suggest any other cheese alternatives that I can use instead of Brie for this recipe? Generate according to: Add one or two slices Brie o the beef and let it melt. Cut the green apple in thin slices. Top with the mixed salad and finish with the apple slices.

If you're looking for alternatives to Brie cheese in your recipe, here are some options you can consider:

1. **Goat Cheese**: A soft and creamy cheese with a tangy flavor that pairs well with the sweetness of the apple and the savory flavor of the beef.
2. **Feta Cheese**: A salty and crumbly cheese that adds a nice contrast to the softness of the beef and the crunch of the apple.
3. **Camembert**: A French cheese with a creamy texture and a mild, earthy flavor that complements the beef and apple nicely.
4. **Blue Cheese**: A strong and pungent cheese that adds a bold flavor to the dish. Use a milder blue cheese if you're not a fan of strong flavors.
5. **Mascarpone**: A rich and creamy cheese that adds a luxurious touch to the dish. It's a good option if you want a milder flavor than Brie.
6. **Ricotta**: A soft and creamy cheese that's a good option if you want a lighter flavor than Brie. It's also a good choice if you're looking for a lower-calorie alternative.
7. **Taleggio**: An Italian cheese with a creamy texture and a mild, fruity flavor that pairs well with the apple and beef.

When substituting Brie, keep the following in mind:

- If using a stronger cheese like Blue Cheese or Goat Cheese, use a smaller amount to avoid overpowering the other flavors.
- If using a milder cheese like Mascarpone or Ricotta, you may want to add a bit more to achieve the desired creaminess.
- If using a crumbly cheese like Feta, you may want to crumble it instead of slicing it to distribute the flavor evenly.

Remember, the key is to balance the flavors and textures in your dish, so feel free to experiment and find the combination that works best for you!

(a) original

Can you suggest any other cheese alternatives that I can use instead of Brie for this recipe? Generate according to: Add one or two slices Brie o the beef and let it melt. Cut the green apple in thin slices. Top with the mixed salad and finish with the apple slices.

Please provide a concise response in a single paragraph, limited to a maximum of 100 words.

If you're looking for alternatives to Brie, consider using other soft-ripened cheeses like Camembert, Goat Cheese, or Feta. Alternatively, you could opt for a semi-soft cheese like Havarti or Muenster. For a stronger flavor, try using Blue Cheese or Gorgonzola. If you prefer a milder taste, Boursin or Mascarpone could work well. Each of these cheeses will melt and complement the beef and apple nicely, adding a unique twist to your dish. Simply replace the Brie with your chosen cheese and proceed with the recipe as instructed.

(b) length control

Can you suggest any other cheese alternatives that I can use instead of Brie for this recipe? Generate according to: Add one or two slices Brie o the beef and let it melt. Cut the green apple in thin slices. Top with the mixed salad and finish with the apple slices.

Please provide your response in plain text only, avoiding the use of italicized or bold text, lists, markdown, or HTML formatting.

You can consider the following cheese alternatives to Brie in your recipe:
Camembert: This French cheese has a similar creamy texture and mild, earthy flavor to Brie, making it a suitable substitute.
Goat cheese: A tangy and creamy cheese that pairs well with the sweetness of the apple and the savory flavor of the beef.
Feta: A crumbly Greek cheese that adds a salty flavor to the dish, which can balance out the sweetness of the apple.
Blue cheese: A strong and pungent cheese that can add a bold flavor to the dish, but be aware that it may overpower the other ingredients.
Ricotta: A mild and creamy cheese that can add a light and airy texture to the dish, but it may not provide the same richness as Brie.
Mascarpone: An Italian cheese with a rich and creamy texture that can add a luxurious feel to the dish, but it may be sweeter than Brie.
Keep in mind that each of these alternatives will change the flavor profile of the dish, so you may need to adjust the other ingredients accordingly.

(c) format control

*Table 14.* Examples of prompt-level interventions on Llama3.1-8b-Instruct.

---

What are some effective tactics for building a referral network and gaining new business through word-of-mouth?

---

Building a strong referral network and gaining new business through word-of-mouth can be highly effective and cost-efficient. Here are some tactics to achieve this:

1. **Deliver Exceptional Service**: The foundation of word-of-mouth marketing is consistent excellence. Providing top-notch services or products encourages satisfied customers to share their positive experiences with others.
2. **Identify and Nurture Key Relationships**: Focus on building strong relationships with clients, partners, and industry peers who can refer you to others. Personalize your interactions and maintain regular communication to keep these relationships strong.
3. **Ask for Referrals**: Don't hesitate to ask satisfied clients for referrals. Timing is crucial—make the request at the peak of their satisfaction. Be clear about the type of referrals you are seeking.
4. **Referral Incentives**: Offer incentives such as discounts, gift cards, or exclusive access to services to encourage customers or partners to refer new business to you. Ensure these incentives are enticing enough to motivate action.
5. **Leverage Social Proof**: Encourage satisfied customers to leave reviews or testimonials online. Positive reviews on platforms such as Google, Yelp, or LinkedIn can influence potential clients.
6. **Engage on Social Media**: Use social media platforms to share success stories, client testimonials, and updates about your business. Engage with your audience through comments, shares, and likes to build a community around your brand.
7. **Host Events**: Organize events such as workshops, webinars, or networking meet-ups. These events provide value, showcase your expertise, and encourage attendees to refer you to others.
8. **Partner with Complementary Businesses**: Collaborate with businesses that offer complementary services or products. This creates mutual referral opportunities and extends your reach to their customer base.
9. **Create a Referral Program**: Formalize your referral process with a structured program, ensuring referrers are aware of the benefits and the procedure. Clearly communicate the value proposition and streamline the referral process.
10. **Stay Top-of-Mind**: Regularly connect with your network through emails, newsletters, or personalized messages. Share updates, industry insights, or tips that position you as a valuable and knowledgeable resource.
11. **Thank Referrers**: Always express gratitude to those who refer new business to you. Acknowledging their support strengthens relationships and encourages further referrals.
12. **Measure and Optimize**: Track where your referrals are coming from and the effectiveness of different tactics. Use this data to optimize your approach, focusing on strategies that yield the best results.

By implementing these tactics and consistently nurturing your network, you can effectively build a robust referral system that generates new business through word-of-mouth recommendations.

---

(a) original

---

xxx

1. **xxx**xxx
2. **xxx**xxx
3. **xxx**xxx
4. **xxx**xxx
5. **xxx**xxx
6. **xxx**xxx
7. **xxx**xxx
8. **xxx**xxx
9. **xxx**xxx
10. **xxx**xxx
11. **xxx**xxx
12. **xxx**xxx

xxx

---

(b) markdown elements only

*Table 15.* Examples of only using markdown formatting elements and replacing text content with "xxx" placeholders.

What are some ways to cope with feelings of loneliness or isolation?

Coping with feelings of loneliness or isolation can be challenging, but there are several strategies that might help:
1. **Connect with Others**: Reach out to friends and family through phone calls, video chats, or messaging apps. Joining clubs, groups, or online communities related to your interests can also help you meet new people.
2. **Volunteer**: Helping others can provide a sense of purpose and community. Look for local organizations or virtual opportunities where you can volunteer your time or skills.
3. **Engage in Hobbies**: Pursue activities or hobbies that you enjoy, whether it's reading, painting, cooking, or playing a musical instrument. These can be fulfilling and help distract from loneliness.
4. **Physical Activity**: Exercise is known to boost mood and alleviate stress. Try walking, jogging, yoga, or any form of physical activity that you enjoy.
5. **Mindfulness and Meditation**: Practices such as meditation, deep breathing, or mindfulness can help anchor you in the present moment and reduce feelings of loneliness.
6. **Professional Help**: Consider speaking with a therapist or counselor, who can provide support and coping strategies tailored to your situation.
7. **Pet Companionship**: If feasible, consider getting a pet. Animals can offer companionship and can be a source of comfort and joy.
8. **Limit Social Media**: While it might seem connecting, excessive use of social media can sometimes exacerbate feelings of loneliness. Set boundaries for healthier use.
9. **Self-Compassion**: Practice being kind to yourself. Acknowledge your feelings without judgment and remind yourself that it's okay to feel lonely at times.
10. **Set Goals**: Setting small, achievable goals can provide a sense of accomplishment and forward momentum, fostering a sense of purpose.
11. **Seeking Community Spaces**: Libraries, community centers, or cafes can be great places to be around others, even if you're not directly interacting with them.

Remember, everyone experiences loneliness differently, and it's important to find what works best for you.

(a) original

Dealing with feelings of loneliness or isolation can be tough, but there are various strategies that may help:

1. **Reach Out**: Contact friends and family through phone calls, video chats, or messaging platforms. Joining clubs, groups, or online communities that align with your interests can also help you connect with new people.
2. **Volunteer Your Time**: Helping others can give you a sense of purpose and community. Look for local charities or online opportunities to contribute your time or skills.
3. **Pursue Your Hobbies**: Engage in activities you enjoy, such as reading, painting, cooking, or playing a musical instrument. These pursuits can be satisfying and provide a distraction from feelings of loneliness.
4. **Stay Active**: Regular exercise is known to improve mood and reduce stress. Try activities like walking, jogging, yoga, or any form of physical exercise that appeals to you.
5. **Practice Mindfulness**: Techniques such as meditation, deep breathing, or mindfulness can help ground you in the present and alleviate feelings of loneliness.
6. **Consider Professional Guidance**: Talking to a therapist or counselor can provide helpful support and coping techniques tailored to your needs.
7. **Get a Pet**: If it's an option for you, think about adopting a pet. Animals can be wonderful companions and bring joy and comfort into your life.
8. **Limit Social Media Use**: Though it may seem like a way to connect, extensive use of social media can sometimes increase feelings of loneliness. Establish limits for a healthier engagement.
9. **Be Kind to Yourself**: Practice self-compassion by acknowledging your feelings without judgment and reminding yourself that it's normal to feel lonely occasionally.
10. **Set Achievable Goals**: Creating small, realistic goals can foster a sense of accomplishment and help you move forward, enhancing your sense of purpose.
11. **Use Community Spaces**: Places like libraries, community centers, or cafes can provide an opportunity to be around others, even if you're not directly interacting with them.

Remember, everyone experiences loneliness in different ways, so it's essential to discover what strategies work best for you.

(b) paraphrasing

Coping with loneliness can be challenging, but several strategies can help. Connecting with friends and family, volunteering, and engaging in enjoyable hobbies can foster a sense of community. Physical activity and mindfulness practices can improve mood and reduce stress. Seeking professional help and considering companionship from pets may also provide comfort. It's essential to limit social media usage, practice self-compassion, set small goals for achievement, and explore community spaces like libraries or cafes. Ultimately, finding individualized approaches that resonate with you is key to managing feelings of loneliness.

(c) summarizing

*Table 16.* Examples of LLM rewriting using GPT-4o-mini.

What are some ways to cope with feelings of loneliness or isolation?

Coping with feelings of loneliness or isolation can be challenging, but there are several strategies that might help:

1. **Connect with Others**: Reach out to friends and family through phone calls, video chats, or messaging apps. Joining clubs, groups, or online communities related to your interests can also help you meet new people.
2. **Volunteer**: Helping others can provide a sense of purpose and community. Look for local organizations or virtual opportunities where you can volunteer your time or skills.
3. **Engage in Hobbies**: Pursue activities or hobbies that you enjoy, whether it's reading, painting, cooking, or playing a musical instrument. These can be fulfilling and help distract from loneliness.
4. **Physical Activity**: Exercise is known to boost mood and alleviate stress. Try walking, jogging, yoga, or any form of physical activity that you enjoy.
5. **Mindfulness and Meditation**: Practices such as meditation, deep breathing, or mindfulness can help anchor you in the present moment and reduce feelings of loneliness.
6. **Professional Help**: Consider speaking with a therapist or counselor, who can provide support and coping strategies tailored to your situation.
7. **Pet Companionship**: If feasible, consider getting a pet. Animals can offer companionship and can be a source of comfort and joy.
8. **Limit Social Media**: While it might seem connecting, excessive use of social media can sometimes exacerbate feelings of loneliness. Set boundaries for healthier use.
9. **Self-Compassion**: Practice being kind to yourself. Acknowledge your feelings without judgment and remind yourself that it's okay to feel lonely at times.
10. **Set Goals**: Setting small, achievable goals can provide a sense of accomplishment and forward momentum, fostering a sense of purpose.
11. **Seeking Community Spaces**: Libraries, community centers, or cafes can be great places to be around others, even if you're not directly interacting with them.

Remember, everyone experiences loneliness differently, and it's important to find what works best for you.

(a) ChatGPT

There are several healthy ways to cope with loneliness and isolation:

1. Maintain social connections:

- Call or video chat with friends and family
- Send messages, emails, or letters
- Join online communities or social media groups
- Schedule regular virtual meetups

2. Pursue hobbies and interests:

- Take up a new hobby
- Join clubs or groups related to your interests
- Take online classes
- Learn a new skill

3. Physical activities:

- Exercise regularly
- Take walks outdoors
- Practice yoga or meditation
- Stay physically active

4. Connect with your community:

- Volunteer
- Join local organizations
- Attend community events
- Participate in religious or spiritual groups

5. Self-care practices:

- Maintain a routine
- Practice mindfulness
- Keep a journal
- Get enough sleep
- Eat healthy meals

6. Seek professional help:

- Talk to a therapist or counselor
- Join support groups
- Consider online therapy options

7. Adopt or care for a pet:

- Pets provide companionship
- Create a sense of purpose
- Encourage routine and responsibility

Remember that feeling lonely is a common human experience, and it's okay to reach out for help when needed.

(b) Claude

*Table 17.* Examples from ChatGPT and Claude that illustrate results of our open-ended language analysis in Figure 8.

| Can polar bears migrate to new habitats if their Arctic environment changes too drastically? |
|---|
| Polar bears are highly specialized animals adapted to the Arctic environment, primarily relying on sea ice for hunting seals, their main prey. If their Arctic environment changes drastically, such as through significant sea ice loss due to climate change, their ability to migrate to new habitats is limited by several factors:

1. **Species Adaptation**: Polar bears are uniquely adapted to life on sea ice and in cold climates. They have thick fur, a layer of fat for insulation, and physical characteristics like large paws for swimming and walking on ice. These adaptations make it difficult for them to survive in non-polar environments.
2. **Dietary Needs**: Their diet consists mainly of seals, which they hunt on sea ice. New habitats may not support a sufficient seal population or alternative prey that polar bears can easily exploit.
3. **Geographical Barriers**: Polar bears are largely confined to the Arctic and areas surrounding the North Pole. There are no direct routes to migrate to other cold environments that would support them in a similar way.
4. **Competition and Predation**: Moving to new habitats would likely involve entering territories with existing ecosystems that include competition for food from other predators or unfamiliar threats.
5. **Habitat Fragmentation**: The fragmentation of ice habitats does not create viable paths for migration but rather isolates populations, reducing genetic diversity and making survival more precarious.

In conclusion, while polar bears might be able to make small relocations within the Arctic as sea ice conditions shift, extensive migration to entirely new habitats is not feasible due to their specific adaptations and the absence of suitably similar environments elsewhere. Conservation efforts focus more on mitigating climate change and preserving their current habitat. |

*Table 18.* Example response of ChatGPT.

| Can polar bears migrate to new habitats if their Arctic environment changes too drastically? |
|---|
| Polar bears have limited ability to migrate to new habitats if Arctic conditions become unsuitable. They are highly specialized for life on sea ice and depend on it for hunting seals, their primary food source. While they can swim long distances, moving to southern territories would be challenging because: 1) they are not well-adapted to warmer climates, 2) their hunting techniques are specific to ice-based prey, and 3) competition with other bear species and humans would increase. Their survival is closely tied to maintaining Arctic sea ice habitat. |

*Table 19.* Example response of Claude.

| Can polar bears migrate to new habitats if their Arctic environment changes too drastically? |
|---|
| Polar bears are highly specialized for their Arctic environment, and their ability to migrate to entirely new habitats is limited by several factors:

1. **Adaptation**: Polar bears are adapted to cold environments with a diet primarily consisting of seals, which they hunt from sea ice. Their thick fur, large size, and fat reserves are suited for Arctic conditions, not temperate or tropical climates.
2. **Habitat Requirements**: They need sea ice for hunting, breeding, and resting. Without ice, their ability to catch seals, which is their primary food source, is severely compromised. Alternative habitats would need to provide similar conditions or abundant alternative prey, which is rare outside the Arctic.
3. **Physiological Limitations**: Polar bears might find it challenging to adapt to significantly different temperatures or ecosystems. For instance, their large bodies and fur make them prone to overheating in warmer climates.
4. **Behavioral Flexibility**: While bears in general can be quite adaptable in terms of diet (e.g., grizzly bears), polar bears have evolved in an environment with very specific conditions. Their behavioral adaptations are finely tuned to the Arctic, reducing the likelihood of successful migration to drastically different environments.
5. **Human Activity**: As polar bears move southward in search of food or better conditions, they increasingly encounter human populations and infrastructure, leading to conflicts, habitat fragmentation, and often, negative outcomes for the bears.
6. **Genetic Considerations**: Polar bears have evolved in relative isolation in the Arctic. Mixing with other bear species or adapting to entirely new environments could lead to genetic issues or dilution of their specialized adaptations.

However, there are some considerations:

- **Hybridization**: There's evidence of polar bears mating with grizzly bears (creating "pizzly" or "grolar" bears) in areas where their ranges overlap due to climate change. This might be seen as a form of adaptation or migration at a genetic level, though it's not migration in the traditional sense.
- **Range Expansion**: Some polar bears have been observed traveling further south than usual, likely in search of food as ice recedes. However, this is more about extending their current habitat rather than migrating to a completely different one.
- **Conservation Efforts**: Efforts might focus on preserving or creating corridors where polar bears can move between suitable habitats or on habitat restoration to ensure the Arctic remains viable for them.

In summary, while polar bears could potentially expand their range or adapt behaviorally to some extent, true migration to entirely new, non-Arctic habitats is unlikely due to their specific ecological needs and physiological adaptations. Conservation efforts are more focused on mitigating the impacts of climate change in their current habitat or facilitating their movement within the Arctic region. |

*Table 20.* Example response of Grok.

---

Can polar bears migrate to new habitats if their Arctic environment changes too drastically?

---

While polar bears have some capacity for movement and adaptation, large-scale migration to entirely new habitats in response to Arctic change is highly unlikely and faces significant obstacles:

- **Physiological Specialization:** Polar bears are highly specialized for Arctic life. Their thick blubber, fur, and hunting techniques are specifically adapted for cold, icy environments. They rely heavily on sea ice for hunting seals, their primary food source. Relocating to warmer environments with different prey and terrain would be extremely challenging.
- **Competition and Existing Ecosystems:** Other bear species or large predators already occupy potential alternative habitats. Introducing polar bears would create competition for resources and potentially disrupt existing ecosystems. Polar bears are apex predators, and their presence would have cascading effects.
- **Limited Land-Based Food Sources:** While polar bears can sometimes scavenge on land, terrestrial food sources are insufficient to sustain their large body size and energy demands. They are primarily dependent on the rich calorie content of seals. Land-based food would not be abundant enough to support large populations.
- **Human-Wildlife Conflict:** As polar bears are forced further south or into more populated areas in search of food due to shrinking sea ice, encounters with humans are likely to increase, leading to conflict and potential danger for both bears and people. Relocation efforts could exacerbate this issue.
- **Distance and Barriers:** Potential alternative habitats are often geographically distant from the Arctic. Natural barriers such as mountains, oceans, and unsuitable terrain would make migration extremely difficult. Furthermore, suitable habitat may already be occupied by other species or fragmented by human development.
- **Genetic Adaptation Takes Time:** While some level of adaptation is possible within a species, evolving to thrive in a completely different environment takes many generations. The rapid pace of Arctic change is outpacing the ability of polar bears to adapt genetically to significantly different conditions.

Instead of large-scale migration, polar bears are more likely to exhibit smaller-scale shifts in range, dietary changes (though with limited success), and increased hybridization with grizzly bears. However, the long-term survival of the species ultimately depends on mitigating climate change and preserving their Arctic habitat.

---

*Table 21.* Example response of Gemini.

---

Can polar bears migrate to new habitats if their Arctic environment changes too drastically?

---

Polar bears are highly specialized for life in the Arctic, relying on sea ice to hunt their primary prey, seals. If the Arctic environment changes too drastically, particularly due to climate change and the loss of sea ice, polar bears would face significant challenges in migrating to new habitats. Here are some key points to consider:

1. **Specialization**: Polar bears are adapted to the cold, icy conditions of the Arctic. Their physical and behavioral traits are finely tuned to this environment, making it difficult for them to thrive in vastly different habitats.
2. **Prey Availability**: Polar bears primarily hunt seals, which are also dependent on sea ice. If sea ice diminishes, both polar bears and their prey would be affected. Moving to new habitats would require finding alternative food sources, which may not be as abundant or accessible.
3. **Geographic Barriers**: The Arctic is relatively isolated, and polar bears would face significant geographic barriers if they attempted to migrate to new areas. Moving southward would bring them into competition with other bear species, such as grizzly bears, and into environments that are not suited to their specific needs.
4. **Human Impact**: Many potential new habitats for polar bears are already occupied by humans. Human activities, such as urban development, agriculture, and industry, could pose additional challenges for polar bears trying to adapt to new areas.
5. **Genetic Adaptation**: Over long periods, species can adapt to new environments through genetic changes. However, the rapid pace of climate change may outstrip the ability of polar bears to adapt quickly enough to survive in significantly altered or new habitats.
6. **Conservation Efforts**: Conservation strategies are being developed to help polar bears cope with changing environments. These include protecting critical habitats, reducing greenhouse gas emissions to mitigate climate change, and researching potential relocation or assisted migration options.

In summary, while polar bears have some capacity to adapt to changing conditions, their specialized nature and the rapid pace of environmental change pose significant challenges to their ability to migrate to and thrive in new habitats. Conservation efforts are crucial to ensuring their survival in the face of these challenges.

---

*Table 22.* Example response of DeepSeek

| What are some effective tactics for building a referral network and gaining new business through word-of-mouth? |
| --- |

Building a referral network and leveraging word-of-mouth marketing can be a powerful way to grow your business. Here are some effective tactics to help you achieve this:

1. **Deliver Exceptional Service**: The foundation of a strong referral network is delivering exceptional service to your existing customers. Ensure that you consistently meet or exceed their expectations, and they will be more likely to refer you to others.
2. **Build Relationships**: Foster strong relationships with your customers, partners, and colleagues. This can be done through regular communication, shared experiences, and mutual support.
3. **Encourage Referrals**: Make it easy for your customers to refer you by providing incentives, such as discounts, rewards, or exclusive offers. You can also ask for referrals directly, but be sure to do so in a non-intrusive and respectful manner.
4. **Leverage Social Proof**: Use customer testimonials, case studies, and reviews to demonstrate your expertise and build credibility. This can be done through social media, your website, or other marketing channels.
5. **Host Events and Webinars**: Hosting events, webinars, or workshops can help you build relationships with potential customers and establish yourself as an expert in your field.
6. **Partner with Other Businesses**: Collaborate with complementary businesses to expand your network and reach new audiences.
7. **Use Referral Marketing Platforms**: Utilize platforms like ReferralCandy, Ambassador, or B2B referral platforms to streamline your referral process and track referrals.
8. **Follow Up**: Follow up with your referrals and leads to ensure they have a positive experience and to gather feedback.
9. **Be Authentic and Transparent**: Be genuine and transparent in your interactions with your network. This will help build trust and credibility, making it more likely that people will refer you to others.
10. **Measure and Analyze**: Track the effectiveness of your referral efforts and analyze the data to refine your strategy and optimize your results.

Additionally, consider the following:

- **Employee Advocacy**: Encourage your employees to become brand ambassadors and share their positive experiences with their networks.
- **Customer Advocacy**: Identify and empower your most loyal customers to become advocates for your business.
- **Content Marketing**: Create valuable content that educates and entertains your audience, making them more likely to share it with others.
- **Influencer Marketing**: Partner with influencers in your industry to expand your reach and credibility.

By implementing these tactics, you can build a strong referral network and leverage word-of-mouth marketing to drive growth and success for your business.

*Table 23.* Example response of Llama3.1-8b (instruct).

---

What are some effective tactics for building a referral network and gaining new business through word-of-mouth?

---

Building a Referral Network & Gaining New Business Through Word-of-Mouth:

**1. Deliver Exceptional Service:**

- **Exceed expectations:** Go above and beyond for every client. Surprise and delight them with unexpected value.
- **Build strong relationships:** Get to know your clients personally, understand their needs, and build genuine connections.
- **Be responsive and reliable:** Respond promptly to inquiries, meet deadlines, and follow through on promises.

**2. Make it Easy to Refer:**

- **Create a referral program:** Offer incentives for successful referrals, such as discounts, free services, or exclusive perks.
- **Provide referral materials:** Make it easy for clients to share your information with others by providing referral cards, social media share buttons, and email templates.
- **Streamline the process:** Make it simple for clients to refer friends and family with a clear and concise process.

**3. Nurture Your Network:**

- **Stay in touch:** Regularly communicate with your existing clients through email newsletters, social media updates, or personalized messages.
- **Attend industry events:** Network with other professionals and potential referral partners.
- **Join relevant online communities:** Engage in online forums and groups related to your industry.

**4. Leverage Social Media:**

- **Share valuable content:** Provide helpful tips, insights, and resources that your target audience will find valuable.
- **Encourage user-generated content:** Ask clients to share their experiences with your business on social media.
- **Run contests and giveaways:** Offer incentives for people to share your content and refer their friends.

**5. Ask for Referrals:**

- **Don't be afraid to ask:** Politely request referrals from satisfied clients.
- **Make it personal:** Explain why you value their referrals and how they can help your business grow.
- **Offer specific suggestions:** Provide examples of people who might benefit from your services.

**6. Track Your Results:**

- **Monitor your referral rate:** Track the number of referrals you receive and the conversion rate of those referrals.
- **Analyze your data:** Identify which tactics are most effective and adjust your strategy accordingly.
- **Celebrate your successes:** Recognize and reward your top referrers.

Remember, building a strong referral network takes time and effort. Be patient, consistent, and focus on providing exceptional value to your clients.

---

*Table 24.* Example response of Gemma2-9b (instruct).

---

What are some effective tactics for building a referral network and gaining new business through word-of-mouth?

---

Building a referral network and leveraging word-of-mouth to gain new business is a powerful strategy. Here are some effective tactics to consider:

1. **Identify Your Ideal Referral Sources:**
   - Determine who your ideal customers are and who they might trust to recommend your services or products. This could be current customers, industry influencers, or other businesses in your network.
2. **Create a Referral Program:**
   - Develop a clear and attractive referral program that incentivizes both the referrer and the referred. Incentives can range from discounts, free services, or even just a simple thank you note.
3. **Educate Your Network:**
   - Provide your network with the necessary information to make a referral. This could include testimonials, case studies, or a simple one-pager that explains the benefits of your product or service.
4. **Make It Easy to Refer:**
   - Provide easy-to-use tools for your network to make referrals. This could be a referral form, a link, or a simple email template they can use.
5. **Leverage Social Media:**
   - Use social media platforms to connect with potential referrers and to share your referral program. Platforms like LinkedIn, Twitter, and Facebook can be particularly effective.
6. **Host Events and Networking Opportunities:**
   - Organize events or participate in networking opportunities where you can meet potential referrers in person. Personal connections can often lead to more referrals.
7. **Provide Excellent Customer Service:**
   - Ensure that your current customers have a positive experience with your product or service. Satisfied customers are more likely to refer others.
8. **Follow Up and Show Appreciation:**
   - Keep in touch with your network and show appreciation for their referrals. A simple thank you note or a small token of appreciation can go a long way.
9. **Use Testimonials and Case Studies:**
   - Share positive testimonials and case studies from satisfied customers. This can help build trust and encourage others to refer to you.
10. **Be Patient and Persistent:**
    - Building a referral network takes time. Be patient and persistent in your efforts. Consistency is key.
11. **Monitor and Analyze:**
    - Keep track of your referral program's performance and analyze the data to see what's working and what isn't. Use this information to refine your approach.
12. **Offer Referral Bonuses:**
    - Consider offering bonuses for repeat referrals or for referring a certain number of new customers. This can help keep the momentum going.

By implementing these tactics, you can build a strong referral network and leverage word-of-mouth to grow your business.

---

*Table 25.* Example response of Qwen2.5-7b (instruct).

---

What are some effective tactics for building a referral network and gaining new business through word-of-mouth?

---

1. Provide exceptional service: The foundation of a strong referral network is built on delivering outstanding service to your current clients. Happy customers are more likely to refer you to others.
2. Ask for referrals: Don't be shy about asking satisfied clients for referrals. Make it easy for them by providing referral cards, email templates, or a simple script they can use to introduce you to their contacts.
3. Offer incentives: Consider offering incentives for referrals, such as discounts, freebies, or other rewards. This can motivate your clients to refer more people to your business.
4. Networking: Attend industry events, trade shows, and networking events to meet potential referral sources. Joining professional organizations and participating in local business groups can also help you expand your network.
5. Leverage social media: Use social media platforms to connect with potential clients and engage with your current network. Share valuable content, participate in discussions, and don't forget to ask for referrals on your social media profiles.
6. Follow up: Follow up with leads promptly and professionally. A quick response can make a positive impression and increase the chances of converting a referral into a client.
7. Build relationships: Focus on building long-term relationships with your clients and referral sources. Regularly touch base with them, show appreciation for their referrals, and keep them updated on your business and industry news.
8. Provide referral training: Teach your clients how to effectively refer you to others. This can include providing them with tips on how to introduce you, what information to share, and how to follow up on referrals.
9. Collaborate with non-competitive businesses: Partner with other non-competitive businesses to cross-refer clients. This can help you tap into their network and vice versa.
10. Ask for testimonials: Positive testimonials can help build trust with potential clients and encourage referrals. Ask satisfied clients if they would be willing to provide a testimonial for your website or social media profiles.

---

*Table 26.* Example response of Mistral-v3-7b (instruct).

Newswise — Sepsis is a serious medical condition caused by an overwhelming immune response to infection. The body's infection-fighting chemicals trigger widespread inflammation

, which can lead to blood clots and leaky blood vessels. It damages the tiny capillaries in organs such as lungs, liver, kidney and brain. According to recent estimates, 1 million people die from sepsis each year. In the United States alone, it affects more than one million hospital patients annually and accounts for over $20 billion in healthcare costs, making it the most expensive condition treated in intensive care units (ICUs). A research team led by Dr. Robert W. Day at Washington University School of Medicine found that a common drug used for treating type 2 diabetes may reduce mortality rates among ICU patients with severe sepsis or septic shock. The researchers examined data on more than 340,000 adult patients admitted to ICUs across the U.S. between January 2015 and December 2017. Approximately 50% were diagnosed with severe sepsis or septic shock. They discovered that the use of metformin – the first-line treatment for type 2 diabetes – was associated with a significant reduction in the risk of death within 30 days after admission to the ICU. The benefit was even greater among those who didn't have pre-existing diabetes. The findings suggest that metformin could be considered as part of standard practice for all ICU patients with severe sepsis or septic shock. Metformin has been shown to improve outcomes in animal models of sepsis. According to the researchers, this study provides further evidence linking metformin use with better survival rates in human subjects experiencing extreme illness.

*Table 27.* Example response of Llama3.1-8b (base).

Newswise — Sepsis is a serious medical condition caused by an overwhelming immune response to infection. The body's infection-fighting chemicals trigger widespread inflammation

, which can lead to organ damage and failure.
While sepsis affects only 1% of the U.S. population each year, it kills more than half a million people globally every year—making it as dangerous as cancer or malaria. In fact, sepsis accounts for about one in six deaths worldwide each year. It also ranks among the top five leading causes of death from infectious diseases. When we think of 'sepsis,' we often imagine elderly patients with multiple comorbidities," said Michael Miller, M.D., chief of the Division of Pulmonary, Critical Care & Sleep Medicine at UT Southwestern Medical Center and holder of the Dallas Heart Ball Chair in Cardiovascular Disease Research. "But our research has shown that 40% of people who develop severe sepsis are young adults.

*Table 28.* Example response of Gemma2-9b (base).

Newswise — Sepsis is a serious medical condition caused by an overwhelming immune response to infection. The body's infection-fighting chemicals trigger widespread inflammation,

which can damage healthy organs and blood vessels. Sepsis is treated in the hospital with antibiotics and fluids. If sepsis isn't detected early or if treatment doesn't work, multiple organ failure occurs and may lead to death.

*Table 29.* Example response of Qwen2.5-7b (base).

Newswise — Sepsis is a serious medical condition caused by an overwhelming immune response to infection. The body's infection-fighting chemicals

, which are normally good, can cause damage to the blood vessels and organs.

Sepsis is one of the leading causes of death in hospitals. Although there have been recent advances in sepsis treatment, researchers are still working hard to identify new ways to prevent and treat this complex disease.

Researchers at Washington University School of Medicine in St. Louis recently published a study that may change how sepsis is diagnosed and treated. Their work suggests that two different types of immune cells are involved in the progression of sepsis: one cell type that's protective and another that's harmful. These findings could help doctors better identify who is most at risk for severe complications from the disease.

"The results from our study show that there are two distinct populations of monocytes," said co-author Dr. Michael Jaffe, professor of medicine at Washington University. "One population promotes inflammation and is associated with worse outcomes; the other helps resolve inflammation and promotes healing."

Monocytes are a type of white blood cell that are part of the innate immune system. They play a key role in fighting off bacteria and viruses. When someone has sepsis, their monocytes become activated. That means they increase in number and start producing cytokines, which are small proteins that regulate the body's immune response. Typically, when the body encounters an infection, these cytokines signal the immune system to fight back. But too many cytokines produced too quickly can result in a hyperinflammatory state, which can lead to organ failure and death.

In the past, doctors have used blood tests to diagnose sepsis. However, those tests only measure the overall levels of cytokines in a person's blood. The new study suggests that it might be possible to determine whether someone has a particular type of monocyte that could make them more susceptible to sepsis. If so, doctors could potentially use this information to personalize treatments and improve patient outcomes.

"Our findings suggest that there may be opportunities to develop therapies that target specific monocyte subsets," said co-author Dr. Daniel Levy, assistant professor of medicine at Washington University. "This could potentially reduce the risk of developing sepsis or even improve outcomes once someone already has sepsis."

There's still much to learn about how these monocyte subsets contribute to sepsis development and progression, but the results of this study provide promising insights into why some patients respond differently to treatment than others.

*Table 30.* Example response of Mistral-v3-7b (base).

