# OpenReview forum: "Idiosyncrasies in Large Language Models"
_ICML.cc/2025/Conference — ICML 2025 poster_

### Official Review · Reviewer_UaTq · 2025-02-28

**Overall Recommendation:** 2

**Summary:**

This paper explores how distinguishable the outputs of different LLMs are from each other. The main results demonstrate that the LLM to produce a given output can easily be predicted by a trained classifier. Further experiments show commonly-used phrases and other idiosyncrasies that differentiate LLMs from one another. Ultimately, the authors conclude that LLM outputs convey deeply-seated differences in their use of language, despite frequently being trained on common data sources.

**Claims And Evidence:**

Claims appear to be supported.

**Essential References Not Discussed:**

None I am aware of

**Experimental Designs Or Analyses:**

- Many results are surprising, such as the high degree of distinguishability between LLMs (Section 3.1) and that the distinguishability is retained even when the outputs are paraphrased or translated by another LLM (Section 4.3).
- Some experiments seem a little superfluous or could be designed to be more interesting. For example, the “Generalization to out-of-distribution responses” in Section 3.1 or the length control aspect of “Prompt-level interventions” experiment in Section 3.2 do not seem to lead to very useful takeaways. Comparatively, it might be interesting to see if the LLMs become less distinguishable when limiting all the LLMs to a restricted, common set of vocabulary during sampling (especially one without their most-commonly used phrases shown in Figure 5). Similarly, the “Sampling methods” experiment in Section 3.2 may instead be more interesting by investigating if the sampling method impacts how recognizable different LLMs are from each other. In other words, repeat the Table 1 main results experiment using top-k sampling, and again for top-p sampling, etc.

**Methods And Evaluation Criteria:**

Most of the results only use one dataset per group of LLMs, as stated in column 2 lines 141-148. These results could change significantly when prompted using a different dataset.

**Other Comments Or Suggestions:**

- Some minor grammatical errors and other issues: for example, the list of LLM types in 3.1 refers to base LLMs mentioned above, when they are the last item in the list
- Figure 4 (and the others of this style) is very useful for results visualization, but looks a little blurry and difficult to read in print.

**Other Strengths And Weaknesses:**

- Some minor grammatical errors and other issues: for example, the list of LLM types in 3.1 refers to base LLMs mentioned above, when they are the last item in the list
- Figure 4 (and the others of this style) is very useful for results visualization, but looks a little blurry and difficult to read in print.

**Questions For Authors:**

Similarly to the point I mentioned earlier, would you be able to share results of how distinguishable the LLMs are from each other (as in Table 1) when using each sampling method?

**Relation To Broader Scientific Literature:**

- The perspective of the paper is different from many other LLM papers, leading to some findings that may be a useful contribution to the LLM research community.
- The “Implications” section does not expand very much on broader implications of the paper, but rather introduces additional experimental results. Moving these additional results to the appendix would allow the authors to better clarify the importance of their work to readers in this section.

**Theoretical Claims:**

N/A

---

> ### Author Rebuttal · Authors · 2025-04-01
>
> We thank the reviewer for the constructive comments. We are happy to address your concerns.
> - **Different prompt datasets**
>
> In our experiments, we primarily use the UltraChat dialogue dataset to generate responses from instruction  LLMs and Chat APIs, and the high-quality pretraining dataset FineWeb for base LLMs. Note that these datasets are already diverse by construction. In addition, in Table 3 of our submission, we have presented results with other prompt datasets (e.g., Cosmopedia, LmsysChat, and WildChat). We find the classifier trained on each of them can achieve >90% accuracy (see diagonal entry). This indicates our results are robust across prompt datasets.
>
> -  **Takeaway of “generalization to OOD responses” and “length control”**
>
> The goal of these two experiments is to study idiosyncratic behavior in controlled experiments. The results of “generalization to OOD responses” show that our classifiers capture robust distinguishable features, indicating  that our observation is general and not dataset specific. The “length control” experiment is to make sure that the high classification accuracy is not due to the short-cut solution which is simply based on the response length.
>
> - **Sampling with restricted, common set of vocabulary (especially without characteristic phrases)**
>
> We restrict the LLMs to a common set of K words during response generation by only sampling relevant tokens composing each word and special character along with special tokens for each LLM. In addition, we do not allow the LLM to output characteristic phrases in Figures 5 and 13 by setting relevant logits to -inf.  Below are the classification results of instruct LLMs using the above sampling strategy:
> | | Llama vs Gemma | Llama vs Qwen | Llama vs Mistral | Gemma vs Qwen | Gemma vs Mistral | Qwen vs Mistral | four way |
> |-----|-----|-----|-----|----- |-----|-----|-----
> |all words | 99.9 | 97.8| 97.0 | 99.9| 99.9|96.1 |96.3|
> |K=2000|98.4| 97.1|99.5 | 99.4 | 99.5 | 98.6|96.9 |
> |K=4000| 99.1 | 97.5|98.5 | 99.4 | 99.6 | 98.0 | 97.1|
>
> The results show that even restricting to a common set of vocabulary, our classifiers can still predict LLM identity with high accuracy.
>
> - **Main results with top-k and top-p sampling**
>
> We use the softmax sampling (T=0.8), top-k sampling (k=100), top-p sampling (p=0.9) to generate responses for instruct LLMs and report their results below:
> | | Llama vs Gemma | Llama vs Qwen | Llama vs Mistral | Gemma vs Qwen | Gemma vs Mistral | Qwen vs Mistral | four way |
> |-----|-----|-----|-----|----- |-----|-----|-----
> |greedy | 99.9 | 97.8| 97.0 | 99.9| 99.9|96.1 |96.3|
> |softmax (T=0.8) | 99.2| 97.3 | 96.1| 99.5| 99.5 |95.7 |95.2|
> |top-k (k=100)  | 99.3| 96.8 |96.2 |99.5 |99.5 | 96.1|95.1 |
> |top-p (p=0.9)| 99.2 | 97.2 | 96.4 |99.6 |99.6 |95.9|95.5|
>
> We observe that using different sampling methods does not alter the results very much (often <2%).
>
> - **Implication too specific and does not expand to broader impact**
>
> We acknowledge that the current implication section contains too many experiments. In the next revision, we will focus more on discussing and highlighting the broader impact of our results in this section. Besides our analysis on synthetic data and model similarity, we also want to mention another implication on privacy and safety. One interesting example would be that the distinguishability of AI models could allow a malicious party to manipulate voting based evaluation leaderboard, e.g., Chatbot Arena. As the identity of the model could be determined from the texts, such an adversary would be able to consistently vote for a desired target model.
>
> - **Grammar errors and LLM type order**
>
> Thank you for pointing this out. We have corrected it in the current version of our paper. We will run a systematic grammar check to proofread the paper.
> -  **Blurry figure**
>
> Thank you for raising this concern. The font size is indeed a little bit small there. We will increase the font size to match that of the figure captions to ensure the results are more readable.

---

### Official Review · Reviewer_RqnH · 2025-03-10

**Overall Recommendation:** 3

**Summary:**

This paper studies the question of how idiosyncratic the responses of different LLMs are. The authors frame this as a classification task where models are trained to predict which LLM (among a fixed set) generated a particular output. The experimental results show that these classifiers have high accuracy, >95% (compared to a base rate of 25%).

The paper then studies specific characteristics that contribute to these idiosyncracies. For example, they find differences in word-level distributions across models, even at the unigram level. They also show that LLM responses can be predicted when responses are rewritten, translated, or summarized. The paper concludes by discussing implications of the results for synthetic data training and using their metrics as a measure of model similarity.

**Claims And Evidence:**

The paper's central claim is that which LLM has written a given text can be easily predicted. In one sense, this claim is very well supported:
- Extensive analysis shows compelling results that LLM outputs are very predictable (when conditioning on up to 4 LLMs).
- Creative range of experiments from different prompts and text shuffling
- The qualitative analysis was nice, focusing on different frequencies of words and letters. The characteristic phrases analysis was also interesting.

However, in another sense, the setting studied by the authors is very narrow with unclear implications:
- The setting -- conditioning on knowing the text is from an LLM and that it's one of 2 or 4 known models -- is contrived. When are we going to come across this in the real-world? As the authors note, this is different from testing whether a response is LLM-generated or not.  Why not a larger classification setting that considers many models?
- The first point in the implications section -- that fine-tuning two different models on the same synthetic data makes their answers harder to distinguish -- is unclear. Why does the data have to be synthetic? This seems like this could be a point about models being trained on different data being easier to differentiate. Wouldn't we have a similar result when models are trained on non-synthetic data? In other words -- why is the synthetic nature of the data important to this claim?
- The second point in the implication section is that the predictability metric can be used as a metric of model similarity. While this is interesting, there isn't much discussion or results about why this would be a useful metric. Is this capturing something that other model similarity metrics aren't capturing? And is there evidence for this?

**Essential References Not Discussed:**

No

**Experimental Designs Or Analyses:**

See responses to "Methods and Evaluation Criteria" and "Other Strengths and Weaknesses"

**Methods And Evaluation Criteria:**

The experimental methodology is generally sound, although some details aren't clearly described. For example, the paper doesn't clearly explain how held-out sets are constructed (e.g. whether responses to the same prompt can appear in both train and test set). See the "Other strengths and weaknesses" section for more discussion. See also the points in "Claims and Evidence"

**Other Comments Or Suggestions:**

See above

**Other Strengths And Weaknesses:**

As mentioned above, the biggest weakness of the paper is the unclear implications. A couple of suggestions for improvement below:
- One possible implication worth exploring is the LLM monoculture literature. How do these results inform our understanding of LLM monoculture, showing e.g. that LLMs make similar mistakes in answering questions or behave similarly when recommending jobs?
- More analysis about what makes pairs of models more predictable would be interesting. E.g. how important is the training data overlap (for models where training data is known)?

The other main drawback of the paper was unclear writing:
- There's not enough detail in the second paragraph of Section 3 to make the experimental setup clear. E.g. how are the held-out sets constructed? Can responses to the same prompt appear in both the train and held-out sets? Does the classifier condition on prompt? What's the total number of prompts in each dataset? These details aren't in the appendix either.
- The tables are very confusing. E.g. what is Table 4 reporting, and what are the units? For many of the tables (e.g. Figure 2, Table 8) it's not clear what's the base rate -- are these all 4-way classification?
- The description of LLMs in lines 125-129 is too strict (i.e. that they "all utilize the Transformer architecture with self-attention" and "trained using an auto-regressive objective"). For example we're beginning to have diffusion-based LLMs ([1] for code specifically, [2] more recently as a general purpose LLM)

[1] Singh, Mukul et al. "CodeFusion: A Pre-trained Diffusion Model for Code Generation".
[2] https://www.inceptionlabs.ai/news

**Questions For Authors:**

See above

=====================

POST-REBUTTAL UPDATE

=====================

I thank the authors for addressing my comments. In response I raised my review.

**Relation To Broader Scientific Literature:**

The paper positions itself well within the literature on dataset bias and human vs. machine-generated text detection. However, it could connect to the LLM monoculture literature -- see "Other Strengths and Weaknesses".

**Theoretical Claims:**

The paper doesn't make substantial theoretical claims requiring proofs.

---

> ### Author Rebuttal · Authors · 2025-04-01
>
> We thank the reviewer for the constructive comments. We are happy to address your concerns.
> - **Contrived setting**
>
> We choose this setting for several reasons:
> 1. We are motivated by prior works [1,2] showing the bias in computer vision datasets and thus adopt a standard classification setup to study the differences of LLMs.
> 2. It helps us evaluate and understand model idiosyncrasies systematically. In section 3 and 4 of our submission, we have designed many analysis experiments around our classification framework, from which we draw many valuable insights.
> 3. Our choice of 4 LLMs per classification group (4 base models together and similarly for instruct models and chat APIs) is mostly for fair comparisons.
>
> We acknowledge that our setup may not consider scenarios where the list of source LLMs could be large and even unknown beforehand. To address this, we conduct a classification experiment with more LLMs (10 in total): ChatGPT, Claude, Grok, Gemini, DeepSeek, Llama, Gemma, Mistral, Qwen and Phi-4, achieving 92.2% accuracy. This indicates our results can be useful in a practical setup with more models. We will include these discussions and new result in our next revision.
> - **Synthetic data**
>
> This result is indeed not specific to synthetic data. We verified fine-tuning LLMs on two non-synthetic datasets yield distinguishable models, e.g., 99.1% for GSM8k versus Math; 98.9% for MMLU versus ARC. We focus on synthetic data in the paper since synthetic data becomes a popular use case of LLM generated outputs. For example, UltraChat dataset consists of mostly GPT-4 responses. Thus we want to highlight this idiosyncratic behavior of training on synthetic data where idiosyncrasies of teacher models can be inherited. We will clarify this point in our next revision.
> - **Model similarity**
>
> Two recent papers have proposed other model similarity metrics, [3] and [4], which we have cited in our submission. Specifically, [3] focused on “vibe”-based similarities based on tones and writing styles. [4] proposed a metric to measure statistical differences between models and their quantized, watermarked or fine-tuned versions. [3]’s method can be extensive and biased as it uses a LLM judge to obtain the vibe scores. [4] is more suitable to detect small changes occurred to one model. Our metric is more general than [3] (not limited to vibes), as we show in Section 4. As opposed to [4], we focus mainly on comparing two different models but not variations of one model.
> - **LLM monoculture**
>
> A recent study that explores LLM monoculture [5] finds that LLM-generated book reviews are more positive than human-written ones. Following their setup, we generate book reviews using Llama-2-7b-chat and Vicuna-13b-v1.5 and find these responses can be classified with near-perfect accuracy. Note that our results are not contradictory with [5] as it is possible to express similar sentiments with different stylistic / lexical choices. Therefore, our work offers a tool to capture idiosyncrasies that characterize LLM monoculture.
> - **Training data overlap**
>
> We added experiments to study the effect of training data overlap. We split GPT-2 pretraining dataset OpenWebText into three parts: a, b and c. We start from training two models on data splits a and b. Then we replace 33% or 66% percent from split a and b with data from split c, which makes the resulting splits overlapping. The classification results on the trained models are:
> |overlapping between a and b|accuracy|
> |-|-
> |0% c|86.4|
> |33% c|86.4|
> |67% c|79.7|
> |100% c|72.6|
>
> As the overlapping increases, the accuracy to classify two trained models is lower, suggesting that pretraining data overlap affects idiosyncrasies.
> - **Experiment setup**
>
> Thanks for pointing this out. We construct the held-out sets by sampling 1K prompts and use the responses generated from these fixed 1k prompts. The prompts for generating the train and held-out sets are separate. We do not include the prompt in the input to the classifier but we find this choice little affects our results. The total number of prompts in UltraChat we sampled from is around 200K. We will add these details in our next revision.
> - **Confusing tables**
>
> In our submission, we clarified at the start of Section 3.2 that “From now on, we report the accuracy of the four-way classification task.”. The base rates would be the results under “original”. We are sorry for the confusion and will make these clearer.
> - **Strict description of LLMs**
>
> We were meaning that we consider Transformer-based LLMs in our paper. We will rephrase this and add related references on diffusion-based LLMs.
>
> [1] A Decade’s Battle on Dataset Bias: Are We There Yet?. ICLR 2025
> [2] Unbiased look at dataset bias. CVPR 2011
> [3] VibeCheck: Discover and Quantify Qualitative Differences in Large Language Models. ICLR 2025
> [4] Model Equality Testing: Which model is this API serving? ICLR 2025
> [5] Generative Monoculture in Large Language Models. ICLR 2025

---

> > ### Comment · Reviewer_RqnH · 2025-04-01
> >
> > I appreciate the rebuttal from the authors. My main concerns have been addressed so I will update my score to 3

---

> > > ### Author Response · Authors · 2025-04-01
> > >
> > > We appreciate your helpful review and discussion about our paper! We are glad that your main concerns have been resolved. We will incorporate above experiment results and your suggestions about writing into the next version of our draft.

---

### Official Review · Reviewer_TS1f · 2025-03-12

**Overall Recommendation:** 3

**Summary:**

This paper shows that deployed large language models have idiosyncracies in their output that makes it possible to distinguish which model generated a particular piece of text. These idiosyncracies seem to transcend the surface structure of the output, persisting after text is shuffled in various ways and even after rewriting.

**Claims And Evidence:**

The primary claims in the paper are well-supported by the results: classification performance is high across the models considered, and even within model families. The authors checked that their classifiers generalize out of distribution and conducted a number of manipulations to the text to try to identify the factors that were making it possible to classify the output of the models. The claims are primarily focused on being able to classify the output well, and the models perform far better than chance although no inferential statistics are presented to support this. It would be good to add error bars to the figures.

**Essential References Not Discussed:**

I did not identify any essential references that were missing.

**Experimental Designs Or Analyses:**

The basic design of the experiments was sound, and the manipulations of the output that were presented in Section 4 were clever and well designed.

**Methods And Evaluation Criteria:**

The methods and evaluation criteria are reasonable for the questions being asked. Some of the methods weren't particularly clear in the main text of the paper. For example, the primary analysis presented on page 3 is based on a classification pipeline that is only explained in the Appendix. Given the centrality of this analysis more details in the paper would be helpful.

**Other Comments Or Suggestions:**

p 3. The order of Base LLMs and Instruct LLMs in the numbered list should be exchanged.

p 6. "Charateristic" -> "Characteristic"

**Other Strengths And Weaknesses:**

The primary weakness of this paper is that it doesn't have a great deal of technical depth -- it primarily uses off the shelf models and classification techniques. However, this isn't the focus of the paper so I don't think this is a major weakness.

**Questions For Authors:**

No questions, in general this was a clearly written paper.

**Relation To Broader Scientific Literature:**

This work situates itself well in the broader literature on large language models, as well as the literature on dataset classification in machine learning. The question that is being asked seemed novel to this paper. I think this question is interesting but not necessarily something that will lead to further technical innovations for large language models -- it seems more motivated by curiosity about the behavior of these models. The most interesting finding was the semantic depth of the idiosyncracies, which suggests that there are relatively deep differences resulting from small changes in architecture and training regimes.

**Theoretical Claims:**

There were no theoretical claims that required assessment -- the primary claims in the paper are empirical.

---

> ### Author Rebuttal · Authors · 2025-04-01
>
> We thank the reviewer for the helpful feedback and recognition of the novelty of the question our paper seeks to answer. We are happy to address your concerns.
> - **The claims are primarily focused on being able to classify the output well, and the models perform far better than chance although no inferential statistics are presented to support this. It would be good to add error bars to the figures.**
>
> Here we report the classification performance on responses from four instruct LLMs, using prompts (10K for train set and 1K for val set) independently sampled from UltraChat three times: 96.3%, 96.1%, 96.0%. The standard deviation in this case is less than 0.2%, which indicates that our reported numbers in the paper are statistically stable.
>
> - **The primary analysis presented on page 3 is based on a classification pipeline that is only explained in the Appendix. Given the centrality of this analysis more details in the paper would be helpful.**
>
>
> Thank you for raising this problem. We agree that moving classification setup in Appendix A to the main paper would be better for readers to understand our methodology. In the next revision of our paper, we will describe these details in Section 3 of the main text.
>
> - **I think this question is interesting but not necessarily something that will lead to further technical innovations for large language models -- it seems more motivated by curiosity about the behavior of these models.**
>
>
> We agree that our work is mostly motivated by curiosity and observation – how current LLMs behave differently in everyday interactions. While our proposed classification framework might not lead to direct technical innovation, it can assist researchers to evaluate and understand differences in LLMs. This is especially important since current frontier models are often not open-sourced (e.g., training data and model weights). Our scientific study can provide several useful insights: as shown in Section 5, our framework can help analyze how synthetic data affects model idiosyncrasies and infer model similarity. These implications may offer valuable insights into current LLMs and, in turn, inform future technical developments.
>
> - **The primary weakness of this paper is that it doesn't have a great deal of technical depth -- it primarily uses off the shelf models and classification techniques.**
>
> Our paper does not focus on proposing new methods but rather focus on characterizing and understanding the differences between LLMs, where we conduct comprehensive analysis to evaluate and understand model idiosyncrasies. Our novelty mostly lies in the results and findings from our carefully designed experiments. Given the growing number of model releases, we hope this work provides valuable insights for both users and model developers.
>
> - **p 3. The order of Base LLMs and Instruct LLMs in the numbered list should be exchanged. p 6. "Charateristic" -> "Characteristic"**
>
> Thank you for pointing out these writing errors. We will correct them in the next revision of our paper.

---

> > ### Comment · Reviewer_TS1f · 2025-04-03
> >
> > Thank you for the response. I appreciate the clarification about the standard errors for the performance. I think we agree on the strengths and weaknesses of the paper and am not changing my score.

---

### Official Review · Reviewer_4tcg · 2025-03-13

**Overall Recommendation:** 4

**Summary:**

This paper studies the samples generated by various LLMs. Specifically, they show that it is possible to effectively determine from which LLM a piece of text was sampled. Furthermore, they connect this predictability to "idiosyncrasies" in the word-level patterns, which persist even when the text has been transformed.

**Claims And Evidence:**

The claims about identification, word-level patterns, persistence under transformation, and the nature of the idiosyncrasies are well-supported by evidence presented by the authors.

**Essential References Not Discussed:**

None

**Experimental Designs Or Analyses:**

Experiments seem broadly sound and effective.

**Methods And Evaluation Criteria:**

The evaluation methods make sense. The authors determine they can train classifiers which work to identify the source LLM for a given text completion and do so with both held out completions and held out prompt datasets. Additionally, the methods for identifying the nature of idiosyncrasies which make these texts predictable seem sound.

**Other Comments Or Suggestions:**

No other comments

**Other Strengths And Weaknesses:**

Strengths:

* The paper is well-written
* Experiments and analysis are thorough and conclusions are well justified
* Experiments for identifying the sampling method as well as showing that predictability persists after transforming the text (e.g. paraphrasing) are interesting

Weaknesses:

* Broader impact may be limited

**Questions For Authors:**

In the paraphrasing experiments, how different is the paraphrased text from the original input text? Also, did you try using other LLMs, aside from GPT-4o mini) for paraphrasing? Were the results consistent if so?

**Relation To Broader Scientific Literature:**

This paper relates to work on identification of LLM-generated text using model-based features (e.g. embeddings) or text features (e.g. n-grams). It also relates more broadly to work which studies the composition of real and synthetic natural language datasets.

**Theoretical Claims:**

There are no proofs in this paper.

---

> ### Author Rebuttal · Authors · 2025-04-01
>
> We thank the reviewer for the positive assessment of our paper and the constructive comments. We are happy to address your concerns.
> - **Broader impact may be limited**
>
> In our submission, we have discussed the broader impact of our results mainly in the first paragraph of the  introduction and Section 5.
> Overall, we believe our results are useful for studying frontier models given that many of their training details are missing in the first place (e.g., training data and model weights). Our results provide a framework that can be used to quantitatively study such differences and reveal the characteristic features of each model. This could potentially help us to build tools for attributing generated text to specific LLMs.
>
> In addition, as described in Section 5, we show that our framework could help study the differences of LLMs when training on synthetic data, and measure model similarity. While synthetic data is a promising direction for scaling training data, our results show that the student model could inherit idiosyncrasies of the teacher model that produce those synthetic data. We further use our framework to show that many AI models are easily classified as ChatGPT while ChatGPT is easily confused with Phi-4 – a model trained with large amounts of synthetic data (Figure 9). These results suggest that our work could help detect the practice of model distillation.
>
> Last, our results can have implications on AI security and safety. One interesting example would be that the distinguishability of AI models could allow adversaries to attack voting based evaluation leaderboard, e.g., ChatbotArena [1]. Specifically, since the identity of the model could be determined from the texts, then an adversary could manipulate the leaderboard by voting consistently for a target model.
>
> We will discuss and highlight these broader impacts of our results in the next revision of our paper.
>
> - **In the paraphrasing experiments, how different is the paraphrased text from the original input text?**
>
> In our submission, we have provided a comparison of the original generated texts and paraphrased texts in Table 15 (page 20) of Appendix. The formatting style remains largely unchanged, e.g., the number of enumerated lists are the same. Most of the differences lie in their word choices, paraphrased texts use different words with similar meanings but do not change the high-level semantic meaning of the original texts. We will add these observations and more text examples in the next revision.
> - **Also, did you try using other LLMs, aside from GPT-4o mini for paraphrasing? Were the results consistent if so?**
>
> Besides GPT-4o-mini, we use Qwen2.5-7b-chat for rewriting LLM responses. Here are the results of classifying the rewritten texts by Qwen2.5-7b-chat on Chat APIs’ responses.
>
> | chat APIs | original | paraphrase | translate | summarize |
> |-|-|-|-|-
> | GPT-4o-mini     | 97.8     | 93.6       | 93.9      | 63.7      |
> | Qwen2.5-7b-chat  | 97.8     | 92.6       | 94.3      | 71.5      |
>
> The results show that our results are robust to the choices of LLMs for rewriting. We will add these results in the next revision.
>
> [1] Chatbot Arena. https://lmarena.ai/

---

> > ### Comment · Reviewer_4tcg · 2025-04-09
> >
> > Thank you for your response and for clarifying those points. I decided to maintain my score of a 4.

---

### Decision · Program_Chairs · 2025-05-01

**Decision:**

Accept (poster)

**Comment:**

This paper shows that deployed large language models have idiosyncracies in their output that makes it possible to distinguish which model generated a particular piece of text. These idiosyncracies seem to transcend the surface structure of the output, persisting after text is shuffled in various ways and even after rewriting.

The reviewers generally agree that the work is well executed and the claims are sound. Several reviewers question the implications of this work and broader impact. The author response addresses these concerns to an extent, and incorporating an improved discussion of these issues will strengthen the paper.